# Unveiling the Unseen: Identifiable Clusters in Trained Depthwise Convolutional Kernels

**Zahra Babaiee**
TU Vienna & MIT
zbabaiee@mit.edu

**Peyman M. Kiasari**
TU Vienna
peyman.kiasari@tuwien.ac.at

**Daniela Rus**
MIT
rus@mit.edu

**Radu Grosu**
TU Vienna
radu.grosu@uwien.ac.at

## Abstract

Recent advances in depthwise-separable convolutional neural networks (DS-CNNs) have led to novel architectures, that surpass the performance of classical CNNs, by a considerable scalability and accuracy margin. This paper reveals another striking property of DS-CNN architectures: discernible and explainable patterns emerge in their trained depthwise convolutional kernels in all layers. Through an extensive analysis of millions of trained filters, with different sizes and from various models, we employed unsupervised clustering with autoencoders, to categorize these filters. Astonishingly, the patterns converged into a few main clusters, each resembling the difference of Gaussian (DoG) functions, and their first and second-order derivatives. Notably, we were able to classify over 95% and 90% of the filters from state-of-the-art ConvNextV2 and ConvNeXt models, respectively. This finding is not merely a technological curiosity; it echoes the foundational models neuroscientists have long proposed for the vision systems of mammals. Our results thus deepen our understanding of the emergent properties of trained DS-CNNs and provide a bridge between artificial and biological visual processing systems. More broadly, they pave the way for more interpretable and biologically-inspired neural network designs in the future.

## 1 Introduction

Convolutional neural networks (CNNs) have been extensively studied in order to understand how they learn ever since their inception. As early as the seminal papers on CNNs, researchers discovered that CNNs learn filters in their initial layers that detect edges or specific edge colors when applied to natural images (Krizhevsky et al., 2012). These learned features bear a strong resemblance to Gabor filters, which are related to responses in the primary visual cortex, and thus present some of the most striking links between neuroscience and machine learning (Goodfellow et al., 2016). However, interpretability significantly decreases as one examines deeper layers. Consequently, subsequent work has mainly focused on inspecting the features learned by convolutional layers, rather than the weights themselves, to elucidate how CNNs operate (Zeiler & Fergus, 2014; Yosinski et al., 2015; Olah et al., 2017; Bau et al., 2017). While studying the learned features by convolutional layers is intuitive, comprehending the filter weights of deep layers of CNNs remains an open question.

In recent years, depthwise-separable convolutional neural networks (DS-CNNs) have become widely adopted in computer vision. The significantly lower computational costs of DS-CNNs enabled MobileNets (Howard et al., 2017) to achieve state-of-the-art accuracy, with substantially fewer parameters and multiplication-addition operations, than traditional CNNs. Following this breakthrough, DS-CNNs have become the de facto standard in most modern CNN architectures, as they offer a favorable option for scaling up networks (Liu et al., 2022). However, most studies analyzing and interpreting the learned kernels and feature maps of CNNs, remained confined to the traditional architectures, using regular convolutions (Zhou et al., 2018; Bau et al., 2017). In contrast, the emergent properties and associated interpretability of DS-CNNs are nowadays largely under-explored.

Through an extensive investigation of several model types and sizes, including regular CNNs and DS-CNNs, trained on ImageNet-1k and ImageNet-21k, we discovered a striking property of DS-CNN kernels. Unlike regular convolutions, depthwise convolutions exhibit repeating patterns in their trained kernel weights. Moreover, these patterns persist throughout all layers, even in deeper

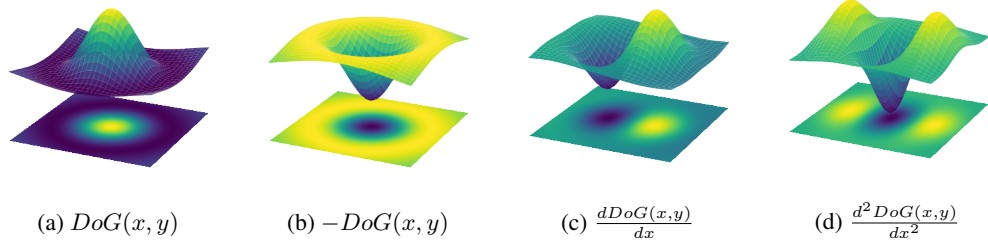

(a) $DoG(x, y)$     (b) $-DoG(x, y)$     (c) $\frac{dDoG(x,y)}{dx}$     (d) $\frac{d^2DoG(x,y)}{dx^2}$

Figure 1: 3D and 2D plots of the DoG function and its derivatives, utilized in Neuroscience for modeling visual receptive fields and in Image Processing for edge and brightness change detection.

stages of the DS-CNNs. This reveals a new level of structure and interpretability in the emergent representations learned by DS-CNNs that has not been identified and characterized before.

Furthermore, we discovered that these identifiable patterns are highly clusterable. To this end, we employed an unsupervised autoencoder-based clustering methodology, mapping each kernel to a single hidden dimension. In the autoencoder outputs, clear clusters emerged, enabling us to categorize nearly all kernels across all layers into eight main classes. Through this classification, we uncovered an intriguing property: the prominent clusters have spatial patterns resembling the difference of Gaussian (DoG) functions and their first and second-order derivatives, as shown in Figure 1.

Fascinatingly, DoG derivatives have been extensively proposed in neuroscience to model biological visual receptive fields (Young, 1987; Young et al., 2001). Incorporating fixed-weight DoG kernels alongside traditional convolutional kernels has shown enhancements in network performance, particularly under conditions of changing lighting and the presence of noise Babaiee et al. (2021). This suggests a profound link between the representations learned by DS-CNNs and those employed in mammalian visual systems. Moreover, the identifiable patterns emerging in DS-CNN kernels have not been previously characterized for modern DS-CNN architectures. Thus, our findings may inform the development of novel bio-inspired network designs and training methodologies. More broadly, this research helps narrow the gap between contemporary machine learning and neuroscience models of sensory processing.

In summary, the key contributions of our work are:

- We conduct the first large-scale analysis of structures emerging in trained DS-CNN kernels.
- We develop unsupervised clustering of millions of DS-CNN filters into core underlying patterns.
- We demonstrate that these discernible patterns are present in all layers of the DS-CNNs.
- We show that the patterns are strikingly similar to the neuroscientific DoG-derivatives models.
- Our findings thus reveal new interpretability and biological parallels of DS-CNNs, respectively.

## 2 RELATED WORK

Our survey covers depthwise separable convolutions (DSC), kernel analysis, and biological vision.

**Depthwise Separable Convolutions.** DSCs decouple the spatial and channel-wise computation in two steps: depthwise convolutions (DC) and pointwise convolutions. This decoupling significantly reduces the computational demands without compromising model efficacy. MobileNet popularized the use of DS-CNNs for efficient, high-performance architectures in resource-limited environments (Howard et al., 2017). In its wake, DS-CNNs have emerged as the cornerstone for a plethora of CNN architectures, including EfficientNet (Tan & Le, 2019a), MobileNetv2 (Howard et al., 2017), MobileNetv3 (Howard et al., 2019), MixNet (Tan & Le, 2019b), and MNasNet (Tan et al., 2019).

**Post-Vit CNNs: A Resurgence.** Leveraging vision transformers' approach (Dosovitskiy et al., 2020), recent studies enhance CNNs with transformer-style and large kernel convolutions. DSCs add efficiency, similar to transformers, for better scalability. ConvNeXt (Liu et al., 2022) extensively analyzes recent vision transformers and presents a highly performant pure convolutional model using 7×7 DCs. Further modernized CNNs have since emerged, including ConvMixers

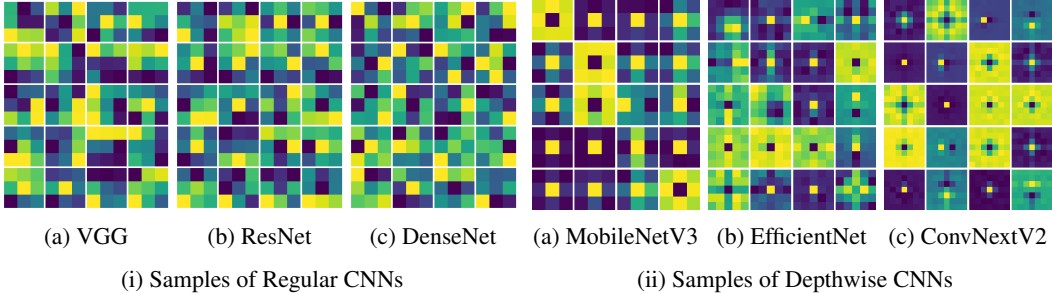

|  (a) VGG | (b) ResNet | (c) DenseNet | (a) MobileNetV3 | (b) EfficientNet | (c) ConvNextV2 |

(i) Samples of Regular CNNs                    (ii) Samples of Depthwise CNNs

Figure 2: Random samples of regular (i) and depthwise (ii) convolutional kernels across all layers. Depthwise convolutions show structured patterns while regular convolutions appear uninterpretable.

which utilize patching and isotropic depthwise blocks (Trockman & Kolter, 2022), Hornets with recursive gated convolutions for high-order spatial interactions (Rao et al., 2022), and MogaNets using dilated DCs (Li et al., 2022). Most recently, ConvNextV2 introduced a fully convolutional masked autoencoder with global-response normalization to enhance inter-channel competition (Woo et al., 2023). Together, these innovations demonstrate that DSCs are a key enabler for scalable, transformer-inspired CNN architectures, which are achieving state-of-the-art results.

**Studies on Trained Convolutional Kernels.**   Studies on trained convolutional kernels have sought to elucidate the learned representations of deep-vision systems. However, most prior work was focused on visualizing kernels in initial layers, where Gabor-like and edge-detecting filters emerge (Krizhevsky et al., 2012). Analyzing deeper layers proved far more challenging, as discernible patterns in kernel weights became increasingly opaque. Thus, many studies centered on visualizing and analyzing learned feature maps rather than the kernels themselves (Olah et al., 2020; Voss et al., 2021). (Gavrikov & Keuper, 2022) assessed distribution shifts between filters across axes like dataset, task, and architecture. Our previous work investigates the presence of on/off-center DoG filters as the two most prominent clusters in depthwise convolutional filters Babaiee et al. (2024). However, the exploration does not extend to other potential clusters in these filters. In this work, we are extending our study covering up to 90-95% of filters clustered, and discovering first and second order derivatives of DoGs in these clusters. Our work thereby spotlights DCs as an avenue to unpacking the black box of trained convolutional networks.

**Biological Models of Vision.**   The neuroscientific investigations have led to mathematical models capturing the response properties of biological vision systems.   (Young, 1987) proposed the Gaussian derivative model, where retinal ganglion and cortical simple cells act as linear filters, well-approximated by Gaussian derivatives. This aligns with difference-of-Gaussians (DoG) models of retinal ganglion receptive fields (Rodieck, 1965). Furthermore, Gabor filters effectively characterize simple cell tuning in the primary visual cortex (V1), but their mathematical formulation requires complex functions (Jones & Palmer, 1987). The same tuning, however, can be achieved with even greater accuracy, by biologically plausible DoG derivatives (Lindeberg, 2021; Tomen et al., 2021). These seminal neuroscience works thus established DoG filters and their directional derivatives, Our findings align with established models of low-level visual processing in mammals, showing parallels between patterns in trained DC kernels and biological vision's receptive field.

## 3    ANALYSIS OF TRAINED KERNELS

We conducted an extensive empirical analysis to compare the patterns emerging in trained convolutional kernels of both regular and DCs, respectively, across diverse state-of-the-art CNN architectures. Our goal was to discern any interpretable patterns in these learned representations. We gathered pre-trained models on ImageNet for the following architectures: AlexNet, VGG, ResNet, DenseNet, MobileNetV2, MobileNetV3, EfficientNet, EfficientNetV2, MixNet, MNasNet, ConvNeXtV1, ConvNextV2, ConvMixer, HorNet, and MogaNet. These architectures use different kernel sizes, and for each model, we used several model sizes and configurations available.

The DS-CNNs analyzed all begin with a regular-convolutional layer, followed by DSC layers. This first layer (also called patching layer), uses a kernel size equal to the stride, to divide the input into patches in the new generation of DS-CNNs, similar to ViT (Dosovitskiy et al., 2020). As observed

Off-Center $dx^2$    Off-Centre      Off-Centre Cross        Off-Centre $dx$    Off-Centre $dy$

On-Center $dx^2$    On-Centre    On-Centre Cross       On-Centre $dx$    On-Centre $dy$

Figure 3: Reconstructed spectrum of 7x7 kernel filters from the 1D hidden code of the autoencoder model, trained on more than 1 million filters. Each segment of the spectrum is marked with the corresponding cluster label. (See Figure 16 for the architecture of the autoencoder.)

in (Krizhevsky et al., 2012), the patching layer learns Gabor-like filters. We therefore omitted it from our analysis and focused on the subsequent DC layers, which are far less studied, up to the deepest stages. This allows us to rigorously characterize the patterns emerging in trained DC kernels across a wide range of layers, from early features in the first layers to late abstractions in the last layers.

To investigate the patterns in trained kernels, we first visually inspected the filters across layers for each model. As shown in Figure 2(i), the regular-convolution filters appear devoid of any discernible structure across models and layers. This aligns with prior analysis that found minimal interpretability beyond early layers in regular CNNs. In contrast, the DC kernels shown in Figure 2(ii), exhibit clear visual patterns that are consistent across diverse model sizes and layers. These patterns are consistent with DoG derivatives as shown in Figure 1. The patterns persist even in the deepest layers, indicating that interpretable representations emerge throughout the full network. Moreover, filters of different kernel sizes converge to similar patterns, suggesting a common underlying structure.

These visualizations reveal a stark difference in the emergent patterns of regular versus DC representations. The recurring interpretable patterns in DCs are explored next through a quantitative analysis. We focus on the $7{\times}7$ kernels from ConvNeXt, which were vectorized and centered before visualization, apply a principal component analysis (PCA) on the kernels across layers. We derive the first 3 PCs explaining the most variance, and we project each kernel onto these 3 components. We then visualize the embeddings in a 3D scatter plot. As shown in Figure 4, the PCA projection reveals distinct clusters forming in the ConvNeXt kernel embedding space. This indicates recurrent identifiable patterns emerging in the learned representations. The recurring filter patterns motivate categorization and further investigation to decode their meaning, as done in the next section.

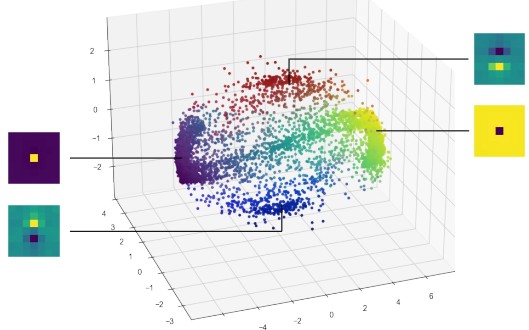

Figure 4: Scatter plot of PCA applied on 7×7 ConvNeXt filters. Sample filters from the 4 most clear clusters are visualized on the side of the plot.

## 4 CLUSTER IDENTIFICATION OF TRAINED KERNELS

### 4.1 MODEL AND METHOD

To carry out a comprehensive classification of the trained kernels, we developed an unsupervised clustering method using autoencoders. The primary objective was to project the kernels onto a compact hidden dimensional space and subsequently execute clustering within this dimensionally reduced space. Distinct models were trained for each kernel size of $5{\times}5$ and $7{\times}7$. For every distinct kernel size, kernels learned from diverse models, and scales exhibiting the corresponding size were assembled. The compilation comprised over one million kernels for each size category. For an extensive enumeration of the models used, please refer to the Appendix.

The autoencoder consists of two main components: an encoder and a decoder. The encoder has four intermediate layers, each followed by a leaky rectified linear unit (Leaky ReLU) activation. The code layer employs a sigmoid activation, to map filters to values within [0,1]. The final decoder

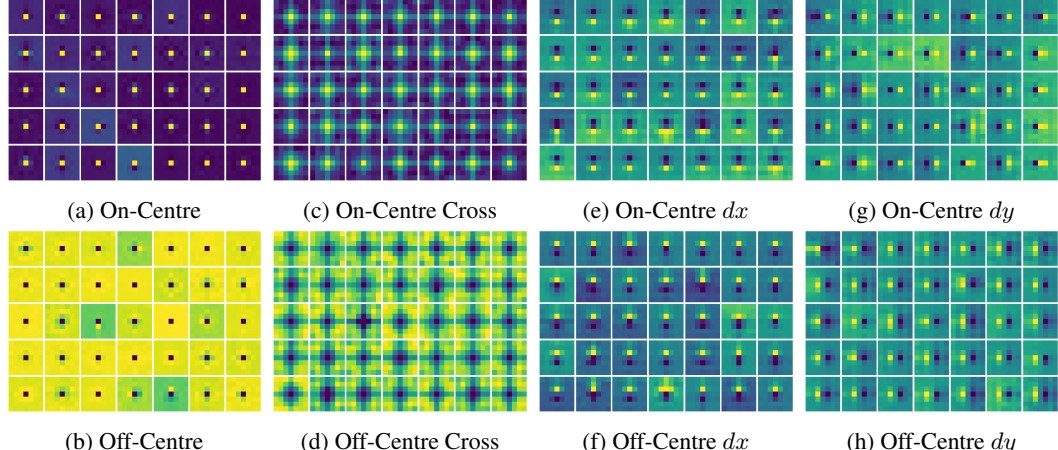

|     |     |     |     |
|-----|-----|-----|-----|
| (a) On-Centre | (c) On-Centre Cross | (e) On-Centre $dx$ | (g) On-Centre $dy$ |
| (b) Off-Centre | (d) Off-Centre Cross | (f) Off-Centre $dx$ | (h) Off-Centre $dy$ |

Figure 5: Random samples from each of the prominent classes of 7×7 kernels of ConvNextV2-tiny trained on ImageNet. Our method classifies the samples with notable accuracy.

layer uses a tanh activation, to accurately reconstruct the original normalized filters in [-1,1]. We utilize a mean-centered cosine similarity loss to accommodate the invariance of filter patterns to linear transformations. Although higher code dimensions yield lower reconstruction loss, we opt for a 1D code to enable convenient visualization and labeling of distinct class intervals for employing the clustering as a classification. While the reconstruction quality is slightly reduced, the interpretability of the clusters is substantially enhanced with this compromise.

Data preprocessing was initiated with the centering of the filters and the normalization of their lengths to unity, ensuring alignment of all filters on a central hyperplane, represented as $\mathbf{1}^T r = 0$. Given the uniform alignment of the normalized filters on a singular hyperplane, dimensionality reduction was possible by transforming the basis of the space to the central hyperplane $\mathbf{1}^T r = 0$.

## 4.2 INFERENCE STAGE

After training the autoencoder, the next step is to identify and label the clusters that emerge, corresponding to different code intervals. To this end, we uniformly sample 500 codes from the 1D space [0,1], with equal spacing, and pass each code through the trained decoder. This generates a spectrum of reconstructed kernels, representing the space of clusters. By visualizing this reconstruction spectrum, we can discern distinct intervals that correspond to different structural patterns.

We manually assign labels to the most prominent clusters, based on their visual patterns.

In Figure 3, we illustrate the reconstruction spectrum for 7×7 ConvNeXt-V2 DC kernels, with labeled intervals corresponding to the 10 most prevalent clusters. The emerging patterns span a diverse range of patterns, as found in the DC kernels. Strikingly, the patterns of DoG functions and their derivatives are clearly discernible, as shown in Figure 1.

Based on the visual motifs detected, we assign semantic labels to each cluster interval. The DoG function DoG(x,y) appears as an On-Centre pattern. Similarly, we label its inverted version $-\mathrm{DoG}(x, y)$ as Off-Centre. Following this nomenclature, we identify the Off-Centre and On-Centre first and second-order derivatives of these DoG functions, respectively, based on their spatial patterns. Interestingly, another common pattern discovered is the shape-of-a-cross pattern, occurring with both on- and off-centers. We refer to them as Off-Centre Cross and On-Centre Cross patterns.

In total, we consistently converge on just 10 core structural patterns spanning DoG functions, their first and second-order derivatives, respectively, and cross-like (2D-absolute-sinc-like) centers and offsets. Next, we study the prevalence and properties of these kernels. Remarkably, through this unsupervised clustering, we are able to categorize millions of heterogeneous depthwise convolution kernels into just a small set of identifiable recurring patterns that arise during training.

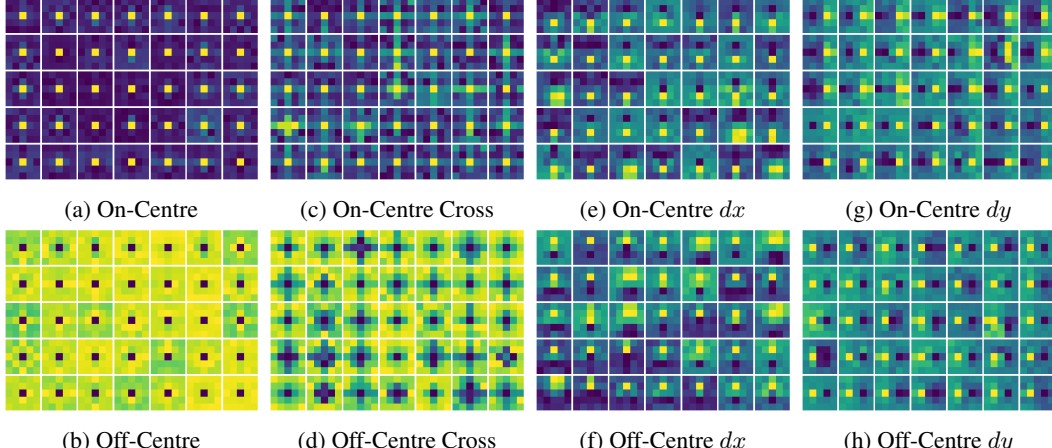

| (a) On-Centre | (c) On-Centre Cross | (e) On-Centre $dx$ | (g) On-Centre $dy$ |

| (b) Off-Centre | (d) Off-Centre Cross | (f) Off-Centre $dx$ | (h) Off-Centre $dy$ |

Figure 6: Random samples from each of the prominent classes of $5{\times}5$ kernels of EfficientNet-b4 trained on ImageNet.

To analyze the learned convolutional filters, we perform inference by sampling 10,000 scalar codes uniformly from $[0, 1]$ and passing them through the decoder. This reconstruction from the latent space helps mitigate potential inconsistencies from the encoder. We then compare each reconstructed filter to its original via cosine dissimilarity, taking the minimum value. If this minimum dissimilarity is below a chosen threshold, we assign the filter to that scalar code's cluster. This clustering based on minimal reconstruction loss allows precise classification of similar filters.

The threshold value is a key parameter controlling cluster assignment and enables reliable clustering. We select a threshold of 0.3 for 7x7 kernels and 0.2 for 5x5 kernels. Using stricter threshold for 5x5 kernels improves robustness because lower dimensional spaces tend to have closer vectors angularly.

Table 1: Percentages of filters in each model that we could classify with high accuracy by using our autoencoder-based clustering method, alongside the model size and model accuracy on ImageNet.

| Model | Parameters | Model Accuracy | Filters Clustered |
|---|---|---|---|
| ConvNextV2_tiny 22k | 28 M | **83.9%** | **98.16%** |
| ConvNextV2_tiny | 28 M | 83.0% | 97.33% |
| ConvNeXt_tiny | 28 M | 82.1 % | 95.21 % |
| HorNet_tiny | 22 M | 82.8% | 80.71% |
| MogaNet_small | 25 M | 83.4% | 78.87% |
| ConvMixer_768_32 | 21 M | 80.1% | 56.64% |
| ConvNextV2_huge_1k_224 | 660 M | 86.3% | 82.08% |
| ConvNextV2_huge_22k_384 | 660 M | 88.7% | 92.41% |
| ConvNextV2_large_1k_224 | 198 M | 84.3% | 90.82% |
| ConvNextV2_large_22k_224 | 198 M | 86.6% | 96.63% |

In Table 1 we show the proportion of filters that we found, from each model, all having 7x7 kernels, and that we were able to successfully classify, with high accuracy. The classification pertains to filters exhibiting a reconstruction loss lower than 0.3. This table, for reasons of succinctness, represents a single model from each unique architecture. A comprehensive table is available in the appendix.

Significantly, the classification was especially successful for the ConvNeXt series. Over 97% of the filters from ConvNextV2 and more than 95% from ConvNeXtv1 were effectively classified, highlighting the efficacy of our methodology in discerning structural patterns within these models.

Moreover, in the MogaNet filters, despite their use of dilated convolutions, a distinct structural pattern also exhibited over 80% classification success. This observation is crucial as it illustrates the ubiquity of the discovered patterns: they emerge in all the DSC-CNN models considered, even if they employ varied convolutional structures, such as dilated DCs.

Furthermore, models with higher accuracy generally had more clusterable filters. The ConvMixer model is interestingly the weakest performer, as it had the most unclustered kernels and somewhat

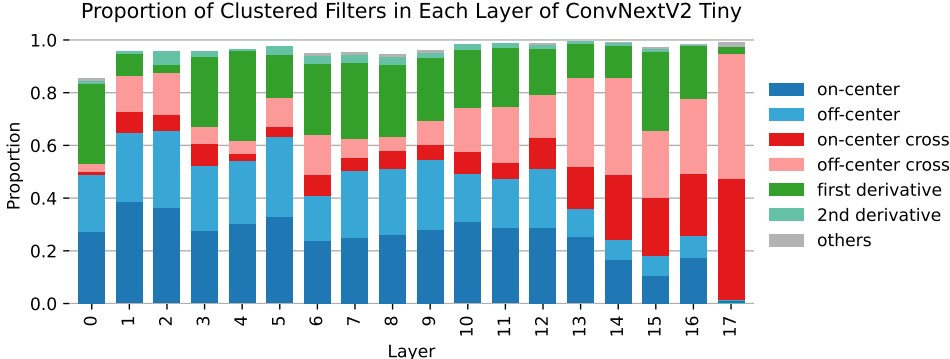

Figure 7: Barplot of relative cluster proportions across all ConvNeXtV2 layers, showing the dominance of DoG cluster patterns across all layers.

noisier weights. Models trained on more data (ImageNet-22k) also exhibited higher clusterability. Overall, these results demonstrate that the recurring patterns we uncovered, arise consistently, especially in high-performing architectures. The depthwise kernel structure becomes increasingly pronounced as models improve, suggesting the patterns are linked to generalization ability.

# 5 UNDERSTANDING THE IDENTIFIED CLUSTERS

In this section, we conduct further in-depth analysis to better understand the properties of the discovered clusters. We study the ConvNeXt-V2 model with 7×7 kernels which has a total of 6624 DC kernels. We selected this model for several reasons. First, larger 7×7 kernels exhibit more diverse and clearer patterns compared to smaller sizes. Second, among 7×7 models, ConvNeXt-V2 demonstrated the cleanest filters with minimal noise and achieved the highest ImageNet accuracy. Third, over 97% of ConvNeXt-V2 kernels were accurately categorized by our clustering approach. By leveraging this state-of-the-art architecture with high clusterability, we can gain key insights into the properties of the recurring patterns uncovered in DC kernels across models. Through both quantitative characterization and qualitative visualization, we unravel the structure of the learned representations in the following. Among the models which have 5×5 kernels, we have selected EfficientNet-b4. This model has a total of 49632 DC kernels.

*Models with $3 \times 3$ Kernels.* We observed that the autoencoder does not fit very well on the kernels with size 3. Our hypothesis is that in low dimensional space, the angles between kernels are smaller and it is harder to disentangle them. Instead, we applied k-means clustering on min-max encoded 3x3 kernels, categorizing them into 10 classes. Visualizations for MobileNetV3 (Appendix Section N) reveal the same recurring On/Off-Center and 1st derivative-like patterns despite using an alternate unsupervised methodology.

To characterize the relative prevalence of each cluster, we compute the proportion of the kernel patterns assigned to each class, across layers, for each model. In Figure 7 we display the barplot visualizing the percentage of ConvNeXt-V2 kernels categorized into each distinct class (pattern). The most frequent patterns are the On-Centre and Off-Centre clusters, followed by their cross variants. For visual clarity, the four first derivative subtypes are merged into one class, which appears next most common. The second derivatives comprise the least frequent group. We label remaining unrecognized patterns as "Others", as they constitute a small fraction. This barplot only includes accurately classified kernels, excluding indiscernible patterns, hence proportions do not total to 1.

We have observed a consistent prevalence hierarchy, maintained across layers, with DoG structures dominating the initial layers. Very interestingly, the proportions of cross-shaped clusters increase in the later layers, while the proportion of DoGs and their first derivatives decreases. In the final convolutional layer, the cross motifs comprise almost all of the most common patterns.

In Figure 11 we display the barplot of the cluster proportions for EfficientNet-B4 kernels, across layers. Similarly to ConvNeXt, the most frequent patterns are the On-Centre and Off-Centre patterns. The first derivative clusters comprise the next most common group. Unlike in ConvNeXt-V2 however, the cross-shaped clusters are far less prevalent in EfficientNet. As shown in Figure 9, we additionally identified two more clusters, named "Square-On" and "Square-Off", that exhibit larger

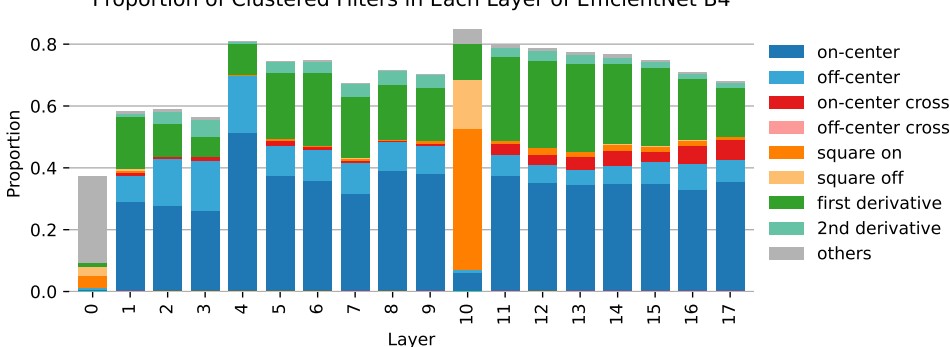

Figure 8: Barplot of relative cluster proportions across all EfficientNet-b4 layers, highlighting the sudden emergence of square kernels in one layer (see appendix Section D).

solid centers, resembling the On-Centre and Off-Centre patterns. As shown in Figure 11, these square clusters uniquely emerge in layer 10. Notably, the first layer in EfficientNet has a lower percentage of correctly classified kernels, with unrecognized "Other" patterns being the most frequent.

The On-Square and Off-Square patterns emerge in almost all models with $5{\times}5$ kernels. Notably, the square shape does not appear centered but rather manifests in the upper left or bottom right corners. We hypothesize this offset localization is due to the small odd size of $5{\times}5$ kernels, where a larger central block would not fit properly. Intriguingly, each model learns these squared clusters fixed to the same corner position for both the On and Off variants. This indicates certain architectural hyperparameters, like kernel size, induce consistent localized deformations in the recurring patterns. We have included more investigations into this in the appendix Section D. Nonetheless, the core identity of the discernible patterns remains intact. Quantifying such subtleties between models provides further insights into the underlying structural representations learned by the DSC-CNNs across a diverse set of network architec-

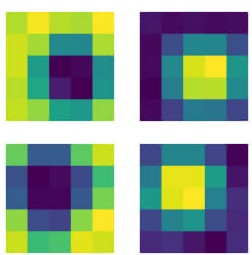

Figure 9: On-Square and Off-Square kernels, emerging only in 5×5 Kernels.

tectures. We have included a more comprehensive set of cluster proportions barplots for additional models in the Appendix.

To further illustrate the consistency of uncovered, learned patterns within each pattern-cluster, we illustrate in Figure 5, random ConvNeXt-V2 kernel samples, drawn from the On-Centre, Off-Centre, Cross, and the first-order derivatives classes. The samples in each category exhibit strong visual similarity to their assigned label, with clean and coherent structures. The fact that thousands of heterogeneous kernels, converge to such (almost) identical motifs is remarkable, and reveals a behavioral simplicity, underlying the emergent representations in depthwise convolutions.

Rather than memorizing a wide range of random patterns, the network distills the filters into a small set of basic building blocks like DoG derivatives and crosses. This provides intuitive insight into how DCs operate: by learning a compact basis set of structural features, that are replicated densely.

To assess total cluster proportions across models, we compared barplots of all models and found notable consistency among versions of each architecture. Figure 11 illustrates this uniformity in models ConvNeXt, ConvNeXtV2, and Hornet, irrespective of model size or training dataset. The full plot containing all models is available in the appendix Section J.1. However, Moganet and the dual sets in ConveNextV2 are exceptions, with Moganet varying first derivative filters and ConveNextV2 showing a shift between on/off-center and cross filters, as detailed in the appendix Section J.2. Training on ImageNet 1K and 21K datasets did not significantly impact these proportions. These observations indicate the architectural design's primary influence over filter distribution, suggesting inherent stability and scalability across these filters.

In order to quantify the emergent patterns, we compute the distribution of total activation (sum of kernel weights) for each cluster. In Figure 10 we show box plots summarizing these distributions. The first-order derivative clusters are centered at zero, indicating balanced positive and negative weights. This distribution is in accordance with the symmetric nature of the derivatives of the

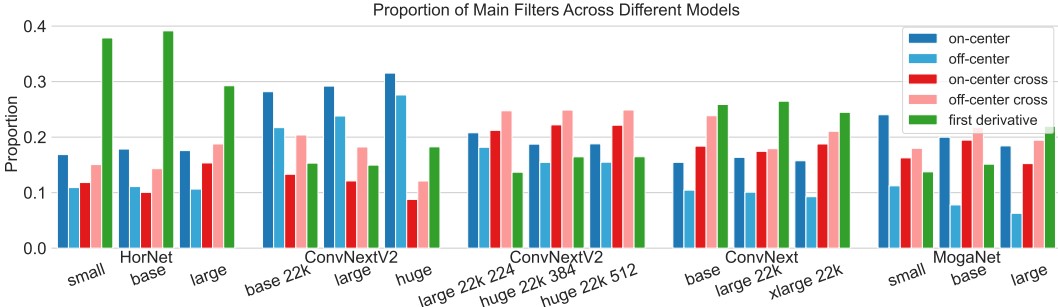

Figure 11: Comparative Barplot of Cluster Proportions Across Models, Showcasing Consistency across architectures, regardless of model size or training data.

Gaussian functions and provides further evidence that these kernels might indeed compute these basis functions.

The cross-shaped clusters exhibit much higher total activations, corresponding to their visual patterns. The On-Centre and Off-Centre clusters show mirrored activation distributions, with the On-Centre DoG kernels having a greater overall weight. These quantitative results reinforce our qualitative observations. The activation statistics capture unique signatures of each pattern that match their assigned visual labels.

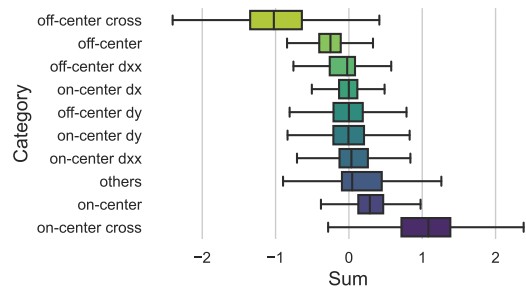

Figure 10: Box plots of total activation for each cluster of ConvNextV2, revealing the symmetry and zero-mean in most clusters.

## 6 CONCLUSION

In our large-scale study, we analyzed patterns in trained DC kernels from various CNN architectures. By visualizing and clustering millions of filters, we identified recurring, interpretable motifs. Notably, the predominant patterns resemble DoGs and their derivatives, akin to models of visual receptive fields in neuroscience.

Our discoveries provide fundamental new insights into the representations learned by modern DS-CNNs. We showed these networks distill dense convolutional filters into a simple vocabulary of basic building blocks, reminiscent of those identified in biological vision. The structural motifs become increasingly pronounced in higher-performing models, suggesting a link between pattern recurrence and generalization capability.

This work bridges the gap between deep learning, neuroscience, and classical image processing. Our approach sets the stage for more interpretable, biologically-inspired convolutional architectures.

**Future Work.** Our study focused on image models. The next steps include analyzing video architectures with 3D convolutions to understand pattern evolution over time, which might align with spatiotemporal visual cortex receptive fields. The identified motifs also lay the groundwork for initialization and regularization methods, aiming to enhance model generalization and efficiency.

The cause of cross-shaped filters is uncertain; a potential link to orthogonal Gaussian function summation is explored in Appendix Section B. Further investigation into these patterns is needed.

Finally, the clusters could inform the development of novel differentiable image filters mimicking the DoG-like learned representations. Integrating these bio-inspired learned kernels into existing CNN operators could lead to enhanced performance and interpretability.

There remain many open questions about the root causes and downstream impacts of the identifiable recurring structures uncovered in depthwise convolutional neural networks. Our discoveries open up numerous avenues for future work, to elucidate the implications of this surprising simplicity, underlying complex emergent deep learning representations.

## 7 ACKNOWLEDGEMENTS

Z.B. is partially supported by the Doctoral College Resilient Embedded Systems, which is run jointly by the TU Wien's Faculty of Informatics and the UAS Technikum Wien. P.K. is supported by the Doctoral Colledge on Trustworthy Autonomous CPS. This research was funded in part, by the Austrian Science Fund (FWF) I 6605.

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

# Appendix

## A  COMPLETE DoG PLOTS

Employing two different first-degree DoGs results in a diverse set of derivatives: four first-order and six second-order derivatives are derived. However, for simplicity and due to the relative rarity of second-derivative filters in our filter set, Figure 12 selectively illustrates only two of these second-order derivatives.

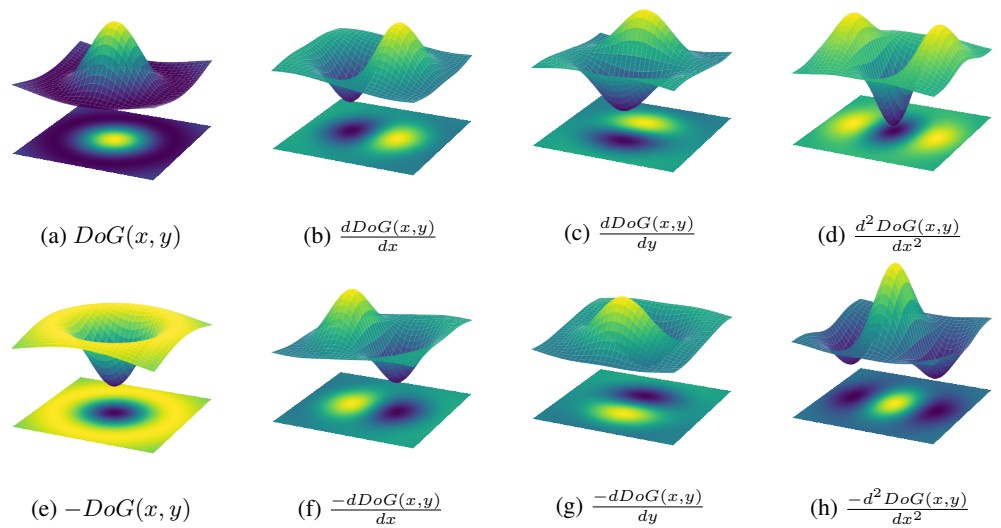

(a) $DoG(x,y)$  (b) $\frac{dDoG(x,y)}{dx}$  (c) $\frac{dDoG(x,y)}{dy}$  (d) $\frac{d^2DoG(x,y)}{dx^2}$

(e) $-DoG(x,y)$  (f) $\frac{-dDoG(x,y)}{dx}$  (g) $\frac{-dDoG(x,y)}{dy}$  (h) $\frac{-d^2DoG(x,y)}{dx^2}$

Figure 12: Complete diagram of the On and Off DoG filters and their derivatives.

For a more comprehensive understanding, DoG and its associated derivatives have been visualized on a 7x7 grid, as depicted in Figure 13.

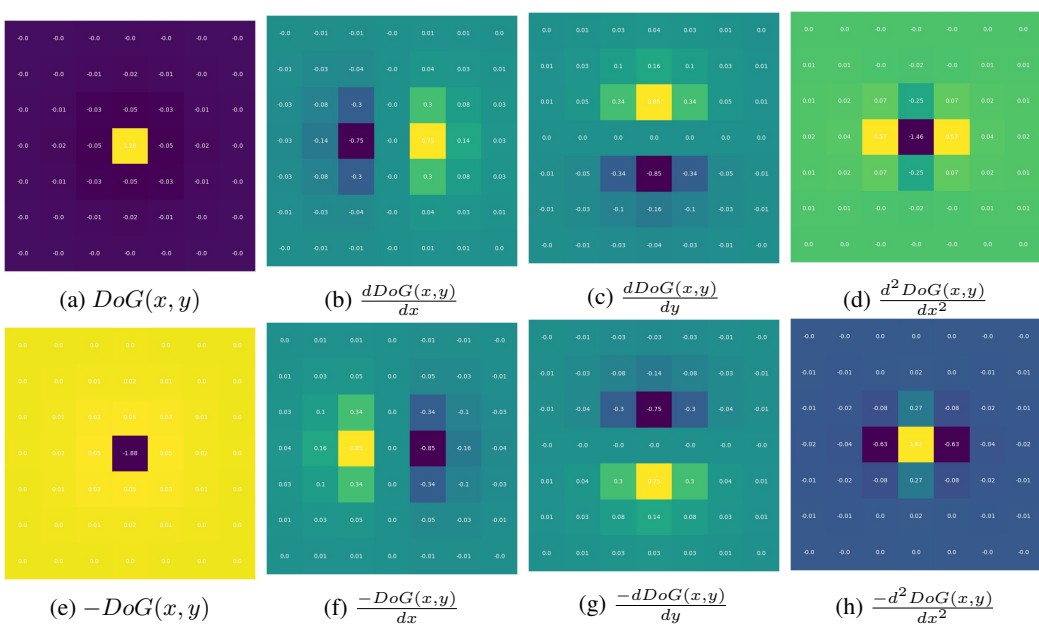

(a) $DoG(x,y)$  (b) $\frac{dDoG(x,y)}{dx}$  (c) $\frac{dDoG(x,y)}{dy}$  (d) $\frac{d^2DoG(x,y)}{dx^2}$

(e) $-DoG(x,y)$  (f) $\frac{-dDoG(x,y)}{dx}$  (g) $\frac{-dDoG(x,y)}{dy}$  (h) $\frac{-d^2DoG(x,y)}{dx^2}$

Figure 13: Complete diagram of the On and Off DoG filters and their derivatives **on a 7x7 grid**.

## B    HYPOTHESIS ON MATHEMATICAL FORMULA OF CROSS-PATTERN FILTERS

The DoGs and their higher derivatives are a well-established concept in image processing. As figure 14 shows, cross-pattern filters share a visual affinity with DoGs, yet their precise mathematical formulation remains unknown to us.

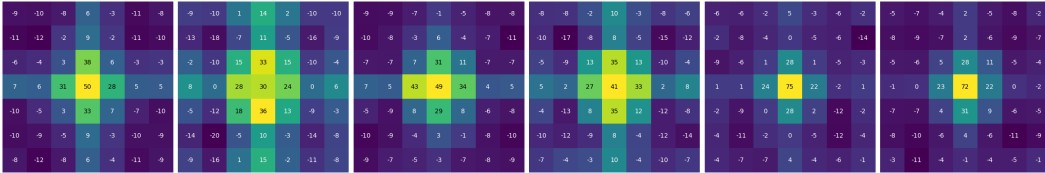

Figure 14: Samples of Cross-Pattern Filters, magnified by 100 to show the first two decimals.

Our investigations suggest that the sum of two orthogonal Gaussian functions provides a near resemblance, as demonstrated in Figure 15, where the functions are visualized on a 7x7 grid across a spectrum of standard deviations:

$$\exp\left(\frac{-x^2}{2\sigma^2}\right) + \exp\left(\frac{-y^2}{2\sigma^2}\right)$$

This formulation is mostly consistent with the spatial characteristics of cross-pattern filters and offers a potential mathematical model for their behavior.

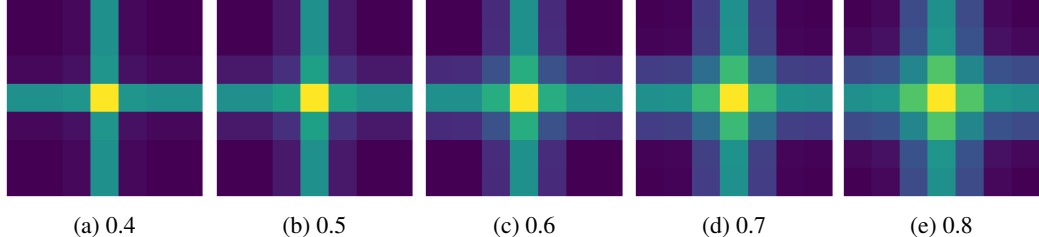

    (a) 0.4        (b) 0.5        (c) 0.6        (d) 0.7        (e) 0.8

Figure 15: Sum of Gaussians with standard deviations ranging from 0.4 to 0.8, on a 7x7 grid.

While this model mirrors the cross-pattern scheme, dissimilarities exist, particularly the more intense central density of the cross-pattern filters, which diverges from the Gaussian summation.

Further research is needed. Future work should refine the model to account for the observed central density and spatial details of the cross-pattern filters, possibly through the use of Gaussian functions.

## C    AUTOENCODER ARCHIRTECTURE

Figure 16 shows the architecture of the autoencoder for kernel size 7, with 1D hidden code in red.

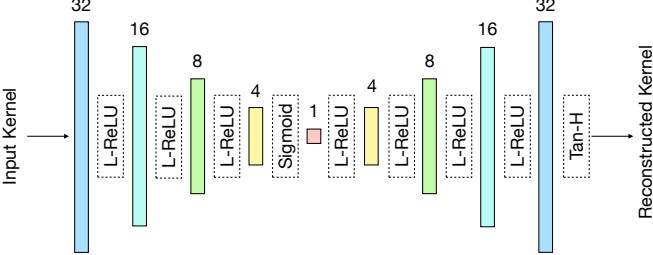

Figure 16: Autoencoder architecture and its layers.

# D    THE CURIOUS CASE OF SQUARE KERNELS

Analysis of the filter clustering proportions in EfficientNets and MobileNets reveals a consistent pattern where one or two layers predominantly learn square kernels in almost all models. For instance, in the EfficientNet-B4 model, this is notably observed in the 22nd layer(layer 10 of 5×5 kernels). To delve deeper into this phenomenon and assess the influence of training procedures and random initialization, we conducted an experiment where the EfficientNet-B4 model was trained using two distinct random seeds over a period of 50 epochs. The resulting barplots, depicted in Figure 18, demonstrate that both training instances exhibit a similar trend to that observed in the Pytorch-released, fully-trained model. Remarkably, in both cases, the 16th layer (layer 4 of 5×5 kernels), shows a higher concentration of clustered filters, mirroring the

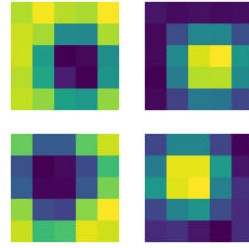

Figure 17: On-Square and Off-Square kernels

pattern in the fully trained model (referenced as Figure 9 in the main text). This trend is particularly intriguing as these layers are positioned at the beginning of a block, suggesting that the model may prioritize training these initial layers, potentially impacting the types of filters learned.

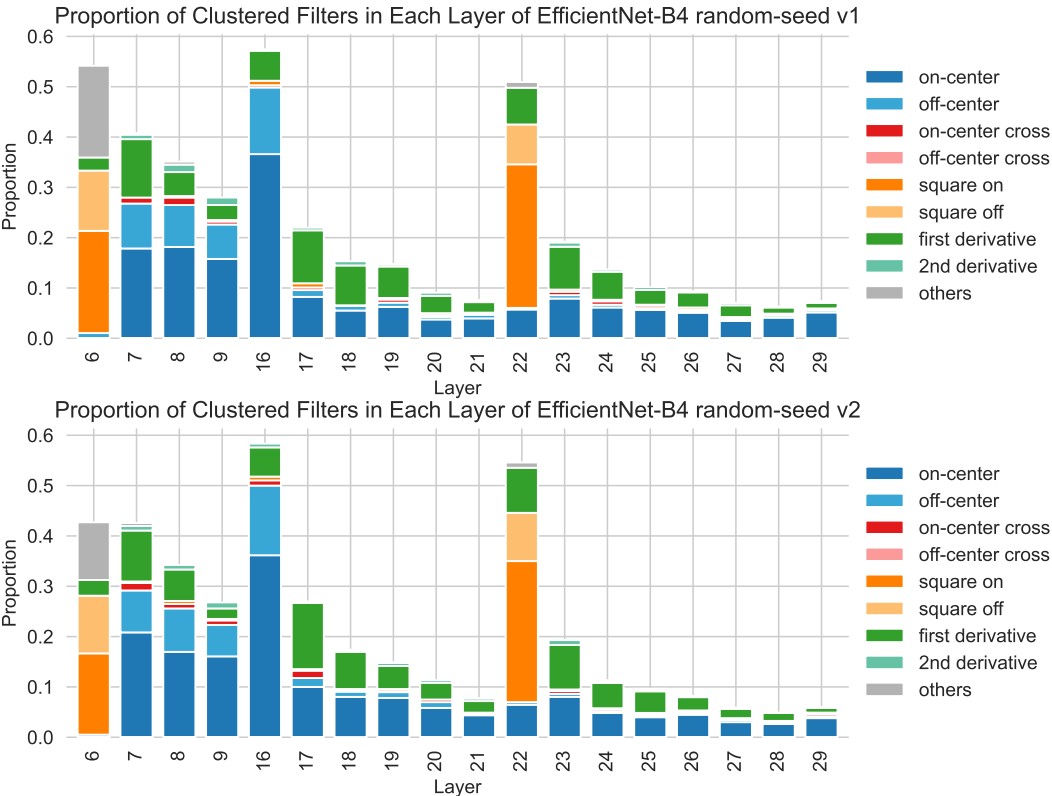

Figure 18: barplot of relative cluster proportions of EfficientNet-B4 after 50 epochs of training with two different random seeds

# E  THE CURIOUS CASE OF HORNET TINY

In contrast to other HorNet models (small, base, and large), HorNet Tiny is illustrated by an abnormal behavior observed in Layer 14. Layer 14 presents an anomaly, with a significant reduction or absence of clustered filters. This deviation is evident in Figure 19.

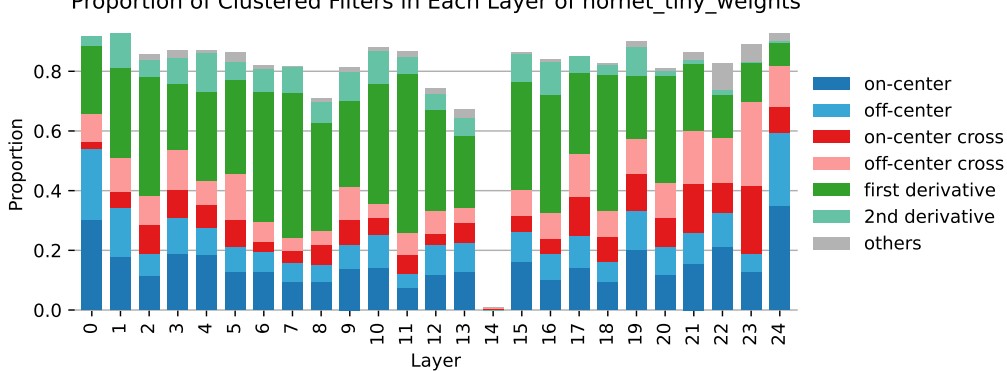

Figure 19: Barplot depicting clustered filter proportions in HorNet Tiny layers, highlighting the unexpected emergence of unidentified filters in Layer 14.

To further clarify this anomaly, we have visualized random samples from Layers 13, 14, and 15 in Figure 20. This comparison distinctly showcases the unusual patterns observed in Layer 14, in contrast to its adjacent layers.

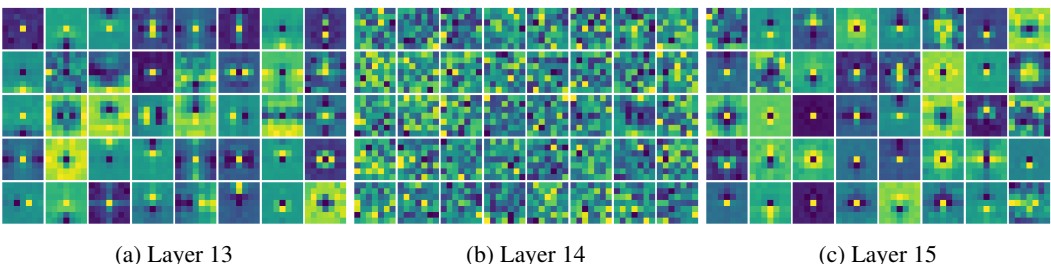

| (a) Layer 13 | (b) Layer 14 | (c) Layer 15 |

Figure 20: Random Samples from Layers 13, 14, and 15 of HorNet Tiny, highlighting the abrupt appearance of seemingly uninterpretable patterns in Layer 14.

Despite investigations, the underlying cause of this phenomenon remains unclear to us. This aspect of our research has yet to be fully understood, presenting an avenue for future exploration.

# F   EXPLORING APPLICATIONS: AN EXPERIMENT ON CONVMIXER

In models utilizing 7x7 filters, the ConvMixer model displayed a notably low score (56%) in filter clustering proportions. Analysis of the model's kernels, as illustrated in Figure 21, revealed the presence of noisy kernels. Despite apparent underlying similarities, many were classified as unclustered by our algorithm, due to the strict loss threshold. We hypothesize these filters are inclined to align with DoG cluster patterns.

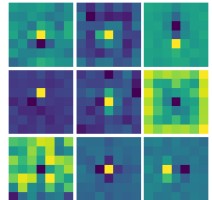

Figure 21: Samples of convMixer filters showing noisy features.

To investigate our hypothesis, we performed an experiment on network initialization. We chose the ConvMixer768-32 model, utilizing a 14-patch size, and trained it over 50 epochs. A second model was initialized with various DoG functions, using random variances similar to trained kernels. The initialization was uniformly distributed: 45% on-center, 10% off-center, 15% cross, 20% first derivative, and the rest second derivatives. This model was then trained under the same conditions.

The table below summarizes our experiment's findings, showing notable improvements in accuracy and an increased proportion of clustered filters in the DoG-initialized model compared to the baseline. Additionally, consistent layer proportions suggest minimal filter changes during training. Figure 25 illustrates these cluster proportions post-training in the barplot.

**Disclaimer.** This experiment was conducted a single time without optimizing proportions and variances; further refinement could improve results, but that was not our main objective here.

| Model | Initilization | Accuracy (%) | Clustered Filters (%) |
|---|---|---|---|
| ConvMixer 768-32 | Kaiming | 66.35 | 56.65 |
| ConvMixer 768-32 | DoGs | **69.01** | **96.03** |

Table 2: Comparison of Normal and Initialized ConvMixer 768-32

Figure 22 compares random filters from the ConvMixer768-32 at epoch 50, both with and without initialization. It is evident that the initialized filters exhibit much clearer patterns.

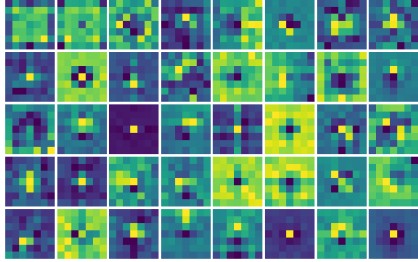

(a) Trained filters with Kaiming initilization

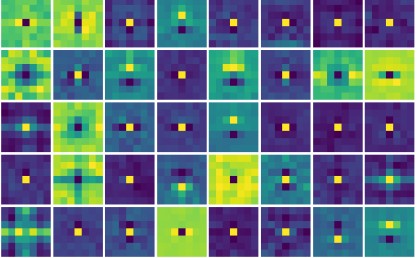

(b) Trained filters with DoGs initilization

Figure 22: Random Samples from ConvMixer768-32 trained without and with DoG initialized weights. Filters with initialization are cleaner and less noisy.

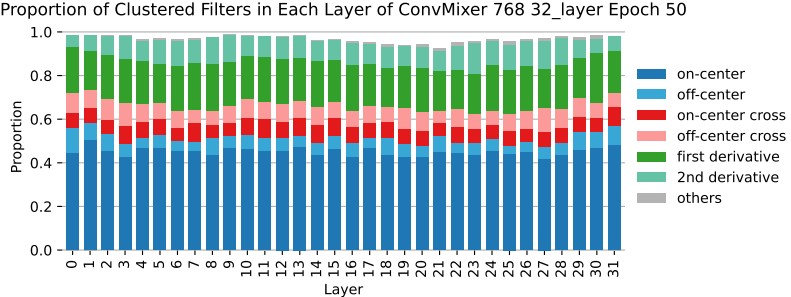

Figure 23: barplot of cluster proportions of ConvMixer initialized by DoGs after training

# G  FILTER PATTERNS TIMELINE

We monitored the Convmixer768-32 model's filters from the initial stage to epoch 50, less than its full 300-epoch training, to observe DoG pattern development. Figure 24 presents barplots of filter proportions at 5, 10, 25, and 50 epochs, showing a gradual emergence of DoG patterns, with the on-center DoG appearing first.

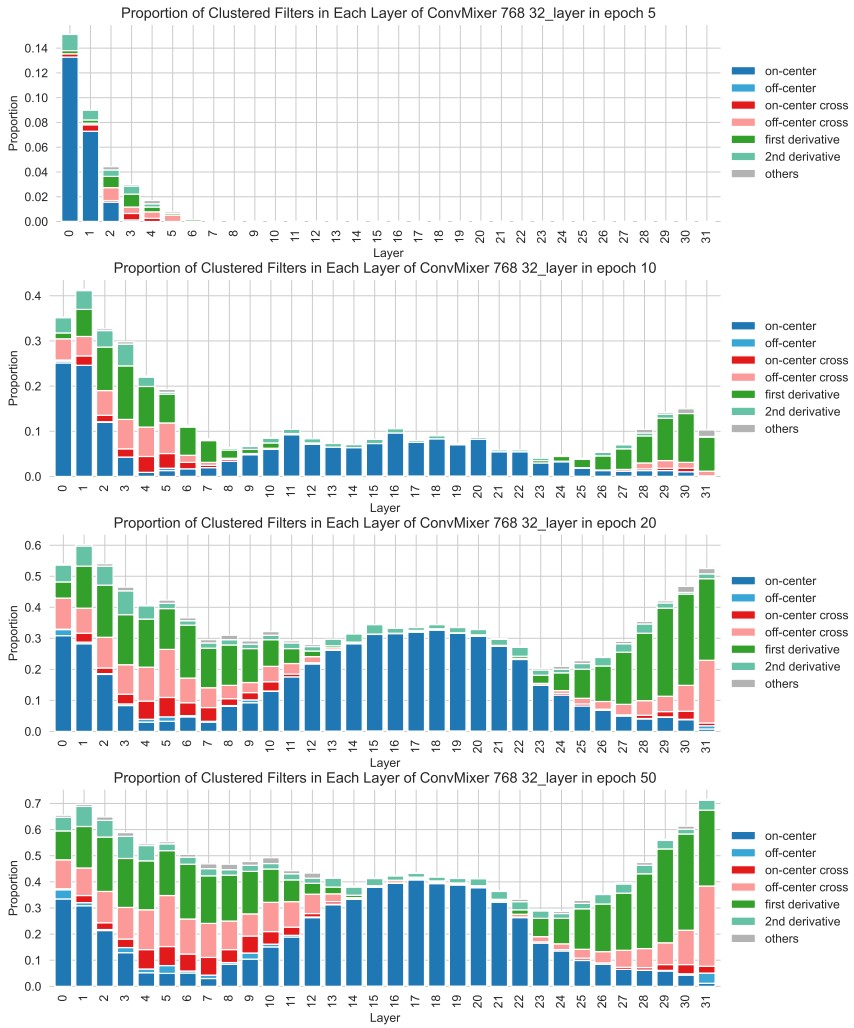

Figure 24: Barplot of relative cluster proportions of ConvMixer768-32 in various epochs.

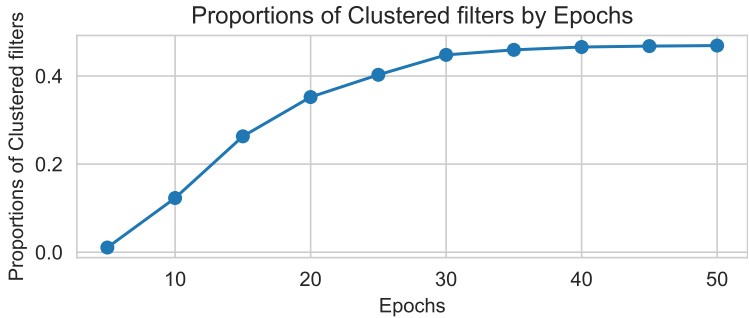

Figure 25: Total proportion of filters clustered across time.

## H GENERALIZATION OR MEMORIZATION?

This section delves into how overfitting impacts the patterns filters learn. We experimented by training the ConvMixer768-32 model on a limited portion of the ImageNet dataset, specifically a subset of just 10 classes. The training was conducted for 200 epochs without any form of regularization, deliberately steering the model towards overfitting. The results of this experiment were quite revealing. Figure 26-b displays a random selection of kernels from the overfitted model. A striking difference is observed when compared to the model trained on the complete ImageNet dataset (Figure 26-a): the kernels lack clear patterns. This finding hints at a potential link between the development of DoG patterns and the model's comprehensive understanding of images, rather than mere memorization.

To further probe this hypothesis, we subjected the same model to training on the small-sized Cifar10 dataset. The kernels from this training, as shown in Figure 26-c, similarly lack discernible patterns, reinforcing the notion that a dataset with a larger variety is necessary for the model to develop recognizable patterns in its filters.

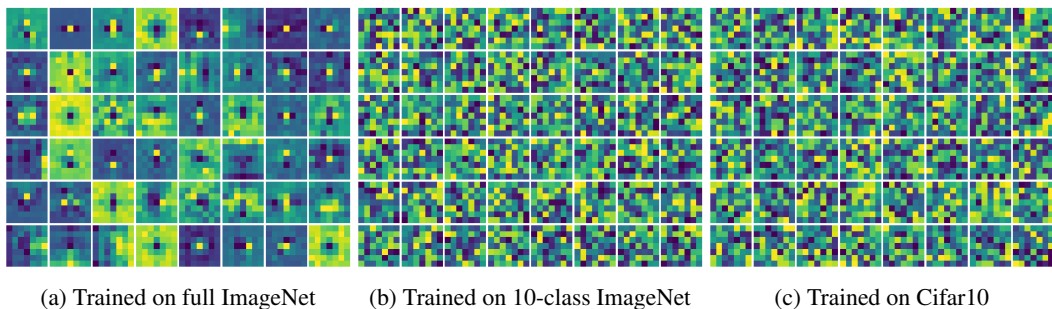

(a) Trained on full ImageNet    (b) Trained on 10-class ImageNet    (c) Trained on Cifar10

Figure 26: Comparative Analysis of Filter Patterns in ConvMixer768-32 Models trained on different data.

## I LARGE 27×27 KERNELS

In order to investigate the presence of the DoG family filters in DS-CNN models with larger kernel sizes, we examined trained filters of model RepLKNet-XL (Ding et al., 2022). Our investigation revealed the presence of DoG-shaped filters in this model, especially in early layers, akin to those observed in smaller models. Figure 27 shows selected samples from the trained kernels of this model. As one can see, despite the large kernels, the patterns emerge in small central area.

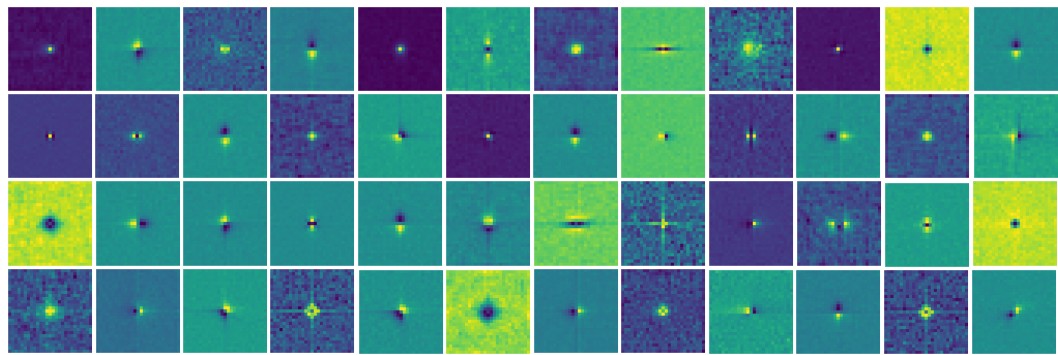

Figure 27: Selected Samples of ReplKNET-XL Model 27×27 Kernels.

## J  RELATIVE CLUSTER PROPORTIONS

### J.1  COMPLETE PROPORTIONS PLOTS

A comprehensive set of bar plots, as referenced in Figure 11, is provided here for detailed analysis.

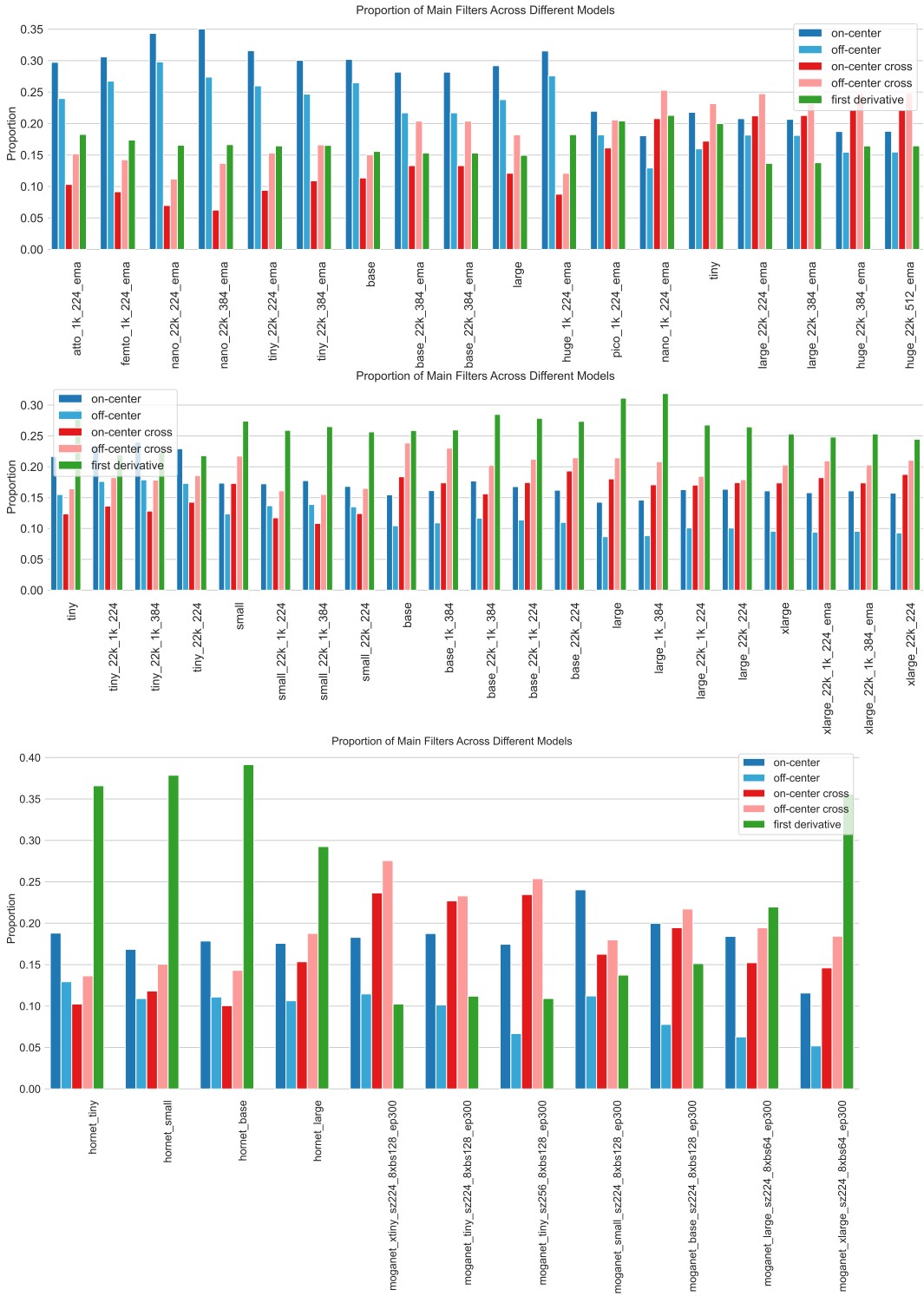

Figure 28: Proportion plots for ConveNextV2, ConveNext, HorNet, and Moganet Models.

## J.2 CONVENEXTV2 DUAL CLUSTER PROPORTIONS

In ConveNextV2 models, a unique duality in cluster proportions is observed, distinct from other models, as Figure 28 reveals two separate proportion sets in ConveNextV2: 11 versions in the first and 7 in the second.

Figure 29 comparing ConveNextV2 huge 1k and 22k suggests the set difference may stem from a shift to on/off-cross from on/off-center filters.

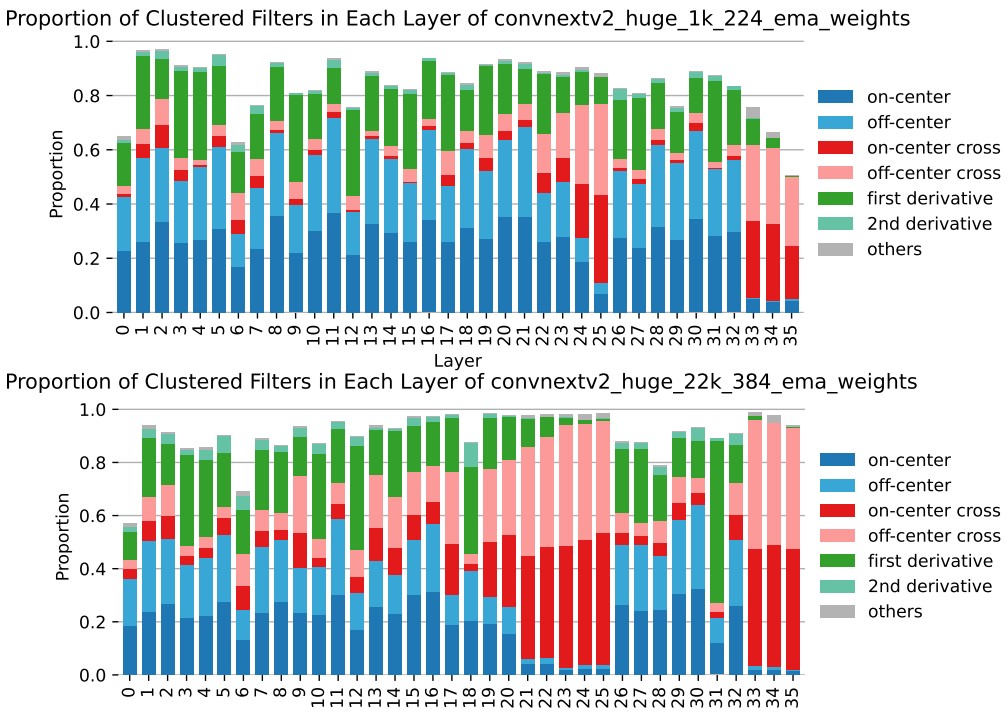

Figure 29: Barplot of relative cluster proportions across all Layers

To validate our hypothesis, we merged labels of on-center and on-cross filters (and off-center with off-cross) in ConveNextV2's proportion barplot. Figure 30 confirms our hypothesis, displaying homogeneous patterns across all proportions. **It is notable that seemingly the merged labels are nearly 50/50 across all models**.

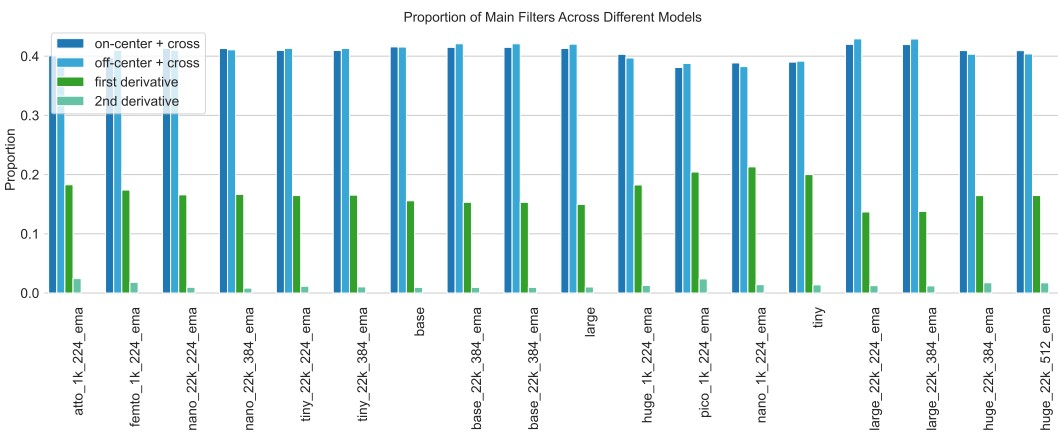

Figure 30: Homogenized Proportions in ConveNextV2 Filters, Merging On-Center/On-Cross and Off-Center/Off-Cross Labels.

## K    THE COMPLETE HIDDEN DIMENSION SPACE RECONSTRUCTION PLOTS

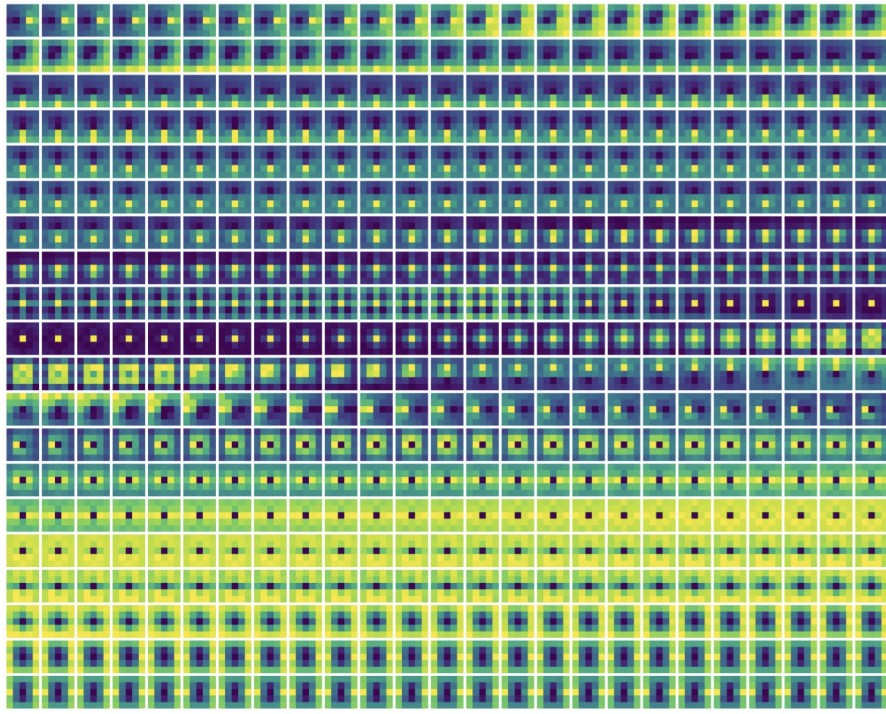

Figure 31: Hidden space spectrum of Autoencoder trained on $5 \times 5$ kernels used for labeling classes.

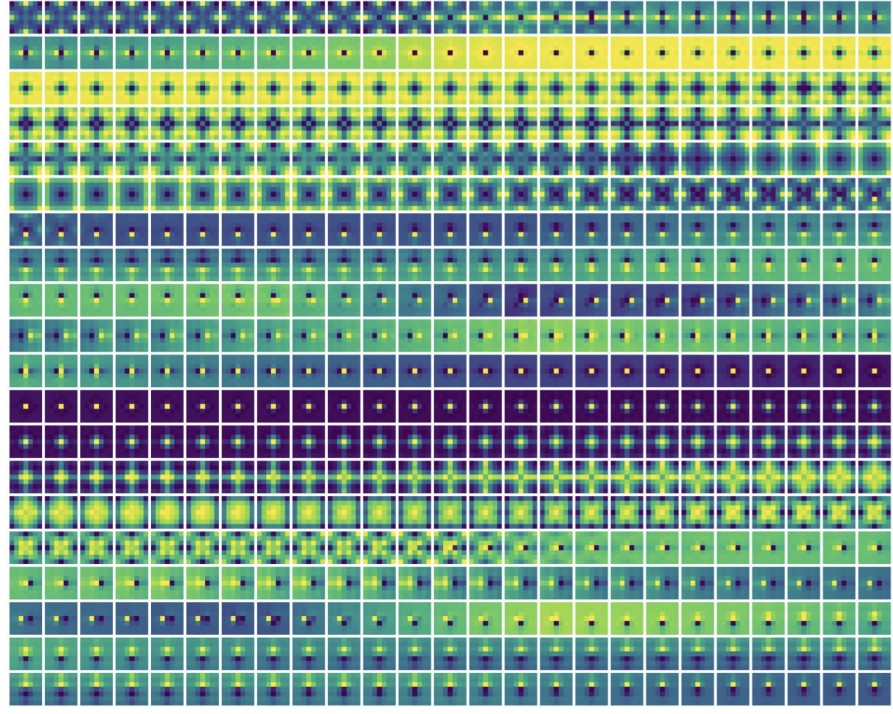

Figure 32: Hidden space spectrum of Autoencoder trained on $7 \times 7$ kernels used for labeling classes.

## L    MODELS USED AND THE PROPORTIONS OF THEIR FILTERS CLUSTERED

| Model | Filters Clustered (%) |
|---|---|
| MobileNetV3 large 100 1k miil 78 0 | 87.09 |
| MobileNetV3 large 100 in21k miil | 85.71 |
| MobileNetV3 large 100 ra | 78.03 |
| MobileNetV3 large 21k | 85.71 |
| MobileNetV3 large | 87.09 |
| MobileNetV3 100 | 78.44 |
| EfficientNet b0 ra | 73.12 |
| EfficientNet b0 | 73.12 |
| EfficientNet b1 | 69.37 |
| EfficientNet b2 ra | 70.48 |
| EfficientNet b3 ra2 | 74.28 |
| EfficientNet b4 ra2 320 | 73.78 |
| EfficientNet el pruned70 | 50.91 |
| EfficientNet el | 52.74 |
| EfficientNet em ra2 | 58.46 |
| EfficientNet es pruned75 | 49.78 |
| EfficientNet es ra | 50.43 |
| EfficientNet lite0 ra | 61.08 |
| tf EfficientNet b0 aa | 76.61 |
| tf EfficientNet b0 ap | 74.97 |
| tf EfficientNet b0 ns | 77.49 |
| tf EfficientNet b1 aa | 73.36 |
| tf EfficientNet b1 ap | 73.83 |
| tf EfficientNet b1 ns | 78.31 |
| tf EfficientNet b2 aa | 71.62 |
| tf EfficientNet b2 ap | 71.81 |
| tf EfficientNet b2 ns | 77.50 |
| tf EfficientNet b3 aa | 70.32 |
| tf EfficientNet b3 ap | 71.48 |
| tf EfficientNet b3 ns | 76.81 |
| tf EfficientNet b4 aa | 67.03 |
| tf EfficientNet b4 ap | 69.67 |
| tf EfficientNet b4 ns | 75.34 |
| tf EfficientNet b5 ap | 69.76 |
| tf EfficientNet b5 ns | 73.32 |
| tf EfficientNet b5 ra | 67.12 |
| tf EfficientNet b6 aa | 63.80 |
| tf EfficientNet b6 ap | 68.15 |
| tf EfficientNet b6 ns | 71.96 |
| MixNet s | 73.25 |
| MixNet m | 66.34 |
| MixNet l | 69.43 |
| MixNet xl ra | 58.91 |
| MnasNet a1 | 71.59 |
| MnasNet b1 | 62.61 |
| SpnasNet 100 | 58.66 |
| FBNet 100 | 59.94 |

Table 3: List of Models with $5 \times 5$ kernels used and their Filter Clustering Results

| Model | Filters Clustered (%) |
|---|---|
| ConvNextV2 atto 224 1k | 96.47 |
| ConvNextV2 femto 224 1k | 97.74 |
| ConvNextV2 pico 224 1k | 95.96 |
| ConvNextV2 nano 224 1k | 96.94 |
| ConvNextV2 nano 224 22k | 97.94 |
| ConvNextV2 nano 384 22k | 97.45 |
| ConvNextV2 tiny 224 1k | 97.33 |
| ConvNextV2 tiny 224 22k | 98.16 |
| ConvNextV2 tiny 384 22k | 98.20 |
| ConvNextV2 base 224 1k | 93.38 |
| ConvNextV2 base 224 22k | 97.56 |
| ConvNextV2 base 384 22k | 97.56 |
| ConvNextV2 large 224 1k | 90.82 |
| ConvNextV2 large 224 22k | 96.63 |
| ConvNextV2 large 384 22k | 96.67 |
| ConvNextV2 huge 224 1k | 82.08 |
| ConvNextV2 huge 384 22k | 92.41 |
| ConvNextV2 huge 512 22k | 92.43 |
| ConvNext tiny 224 1k | 95.21 |
| ConvNext tiny 224 22k_1k | 87.55 |
| ConvNext tiny 384 22k_1k | 86.91 |
| ConvNext tiny 224 22k | 88.09 |
| ConvNext small 224 1k | 94.68 |
| ConvNext small 224 22k_1k | 94.48 |
| ConvNext small 384 22k_1k | 93.84 |
| ConvNext small 224 22k | 94.57 |
| ConvNext base 224 1k | 93.57 |
| ConvNext base 384 1k | 93.13 |
| ConvNext base 224 22k_1k | 93.82 |
| ConvNext base 384 22k_1k | 92.91 |
| ConvNext base 224 22k | 93.87 |
| ConvNext large 224 1k | 91.26 |
| ConvNext large 384 1k | 90.77 |
| ConvNext large 224 22k_1k | 88.59 |
| ConvNext large 384 22k_1k | 88.06 |
| ConvNext large 224 22k | 88.93 |
| ConvNext xlarge 224 1 | 87.67 |
| ConvNext xlarge 224 22k_1k | 87.95 |
| ConvNext xlarge 384 22k_1k | 87.67 |
| ConvNext xlarge 224 22k | 88.11 |
| MogaNet xtiny 224 1k | 78.19 |
| MogaNet tiny 224 1k | 80.82 |
| MogaNet tiny 256 1k | 84.16 |
| MogaNet small 224 1k | 78.87 |
| MogaNet base 224 1k | 70.22 |
| MogaNet large 224 1k | 58.49 |
| MogaNet xlarge 224 1k | 55.30 |
| HorNet tiny 224 1k | 80.71 |
| HorNet small 224 1k | 79.09 |
| HorNet base 224 1k | 72.86 |
| HorNet large 224 1k | 68.58 |
| ConvMixer 512 20_layer 1k | 58.02 |
| ConvMixer 768 32_layer 1k | 56.65 |
| ConvMixer 768 initialized* | 96.03 |

Table 4: List of Models with $7 \times 7$ kernels used and their Filter Clustering Results
*Note: Please see Section F

## M    PROPORTION BAR CHARTS

Proportion of Clustered Filters in Each Layer of moganet_tiny_sz256_8xbs128_ep300_weights

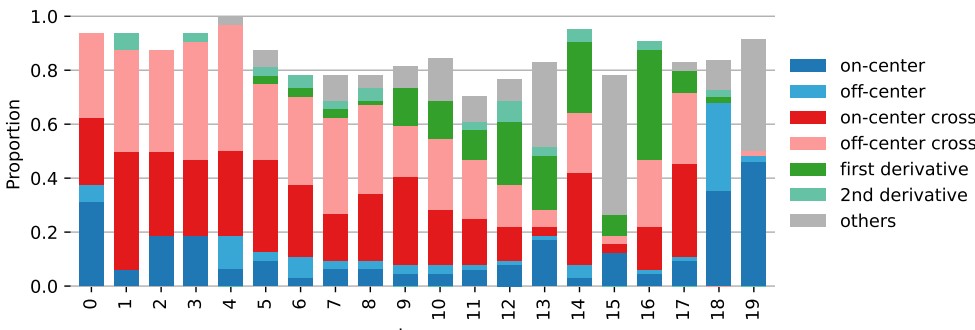

Proportion of Clustered Filters in Each Layer of moganet_base_sz224_8xbs128_ep300_weights

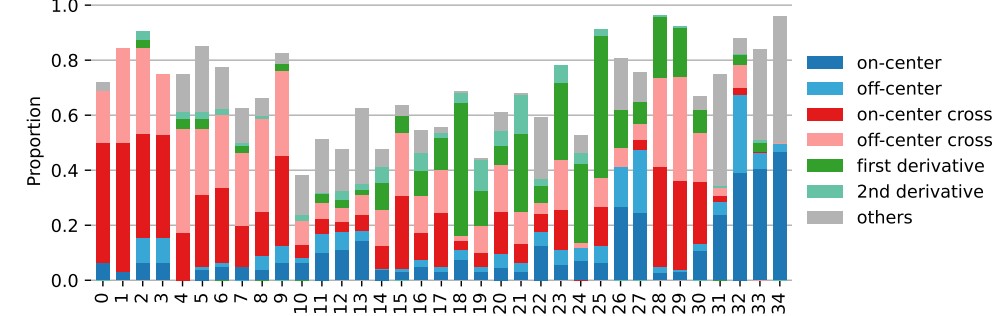

Proportion of Clustered Filters in Each Layer of moganet_large_sz224_8xbs64_ep300_weights

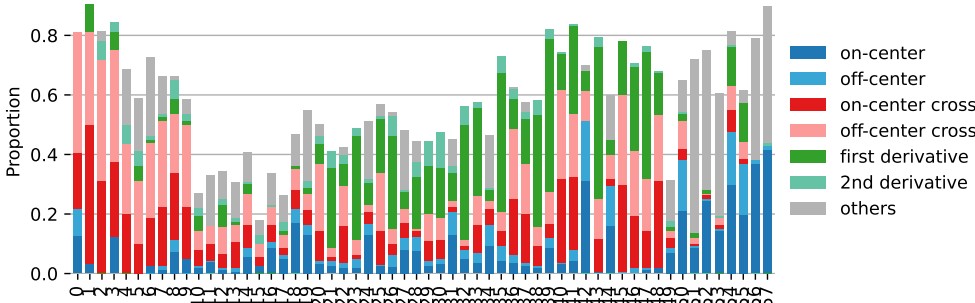

Proportion of Clustered Filters in Each Layer of moganet_xlarge_sz224_8xbs64_ep300_weights

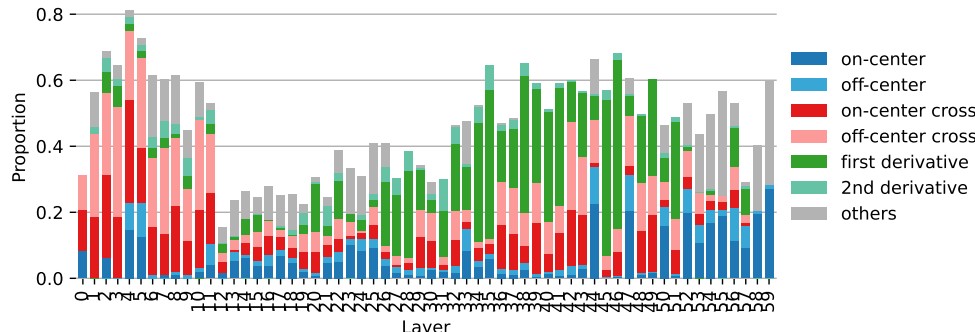

Figure 33: Barplot of relative cluster proportions across all Layers

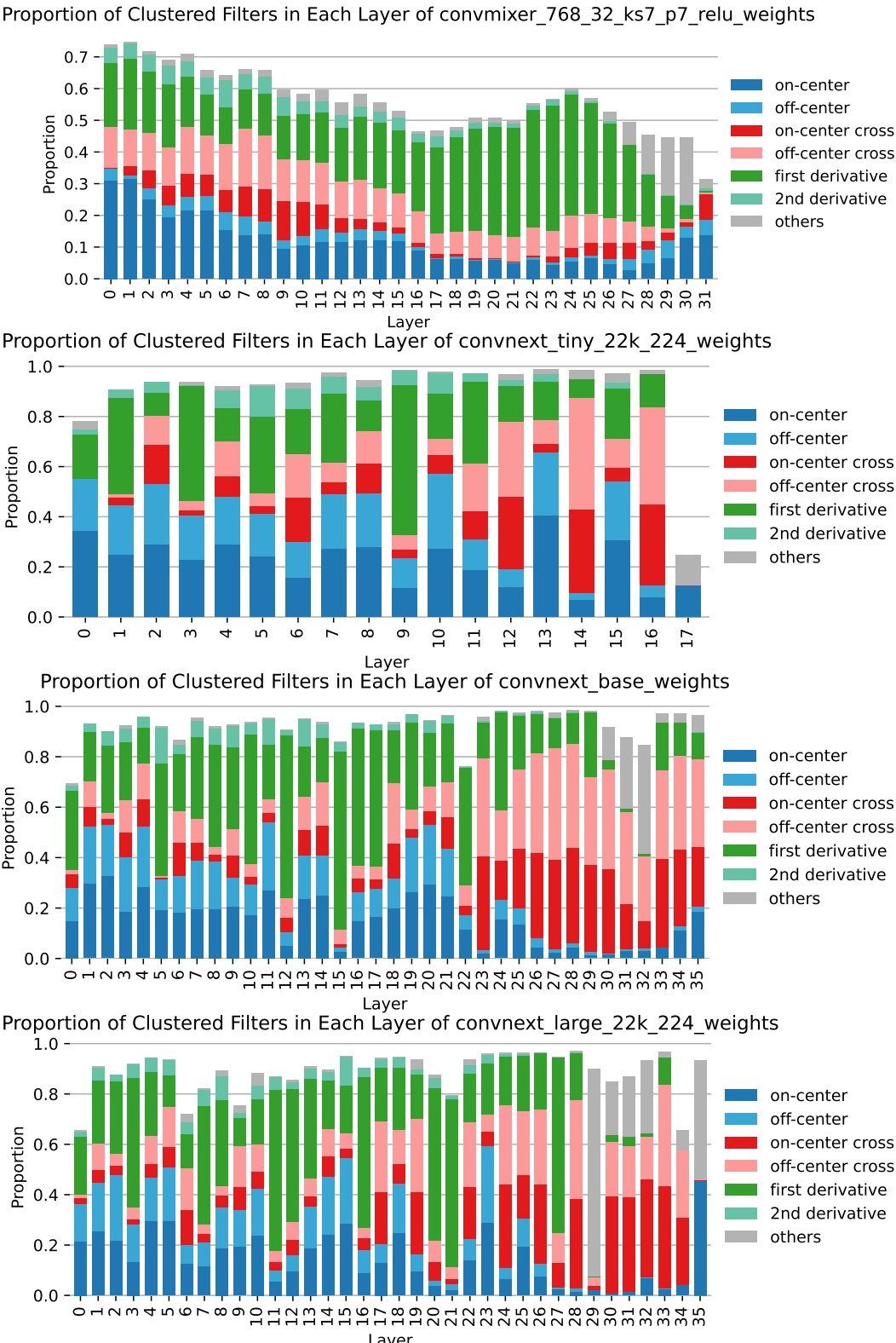

Figure 34: Barplot of relative cluster proportions across all Layers

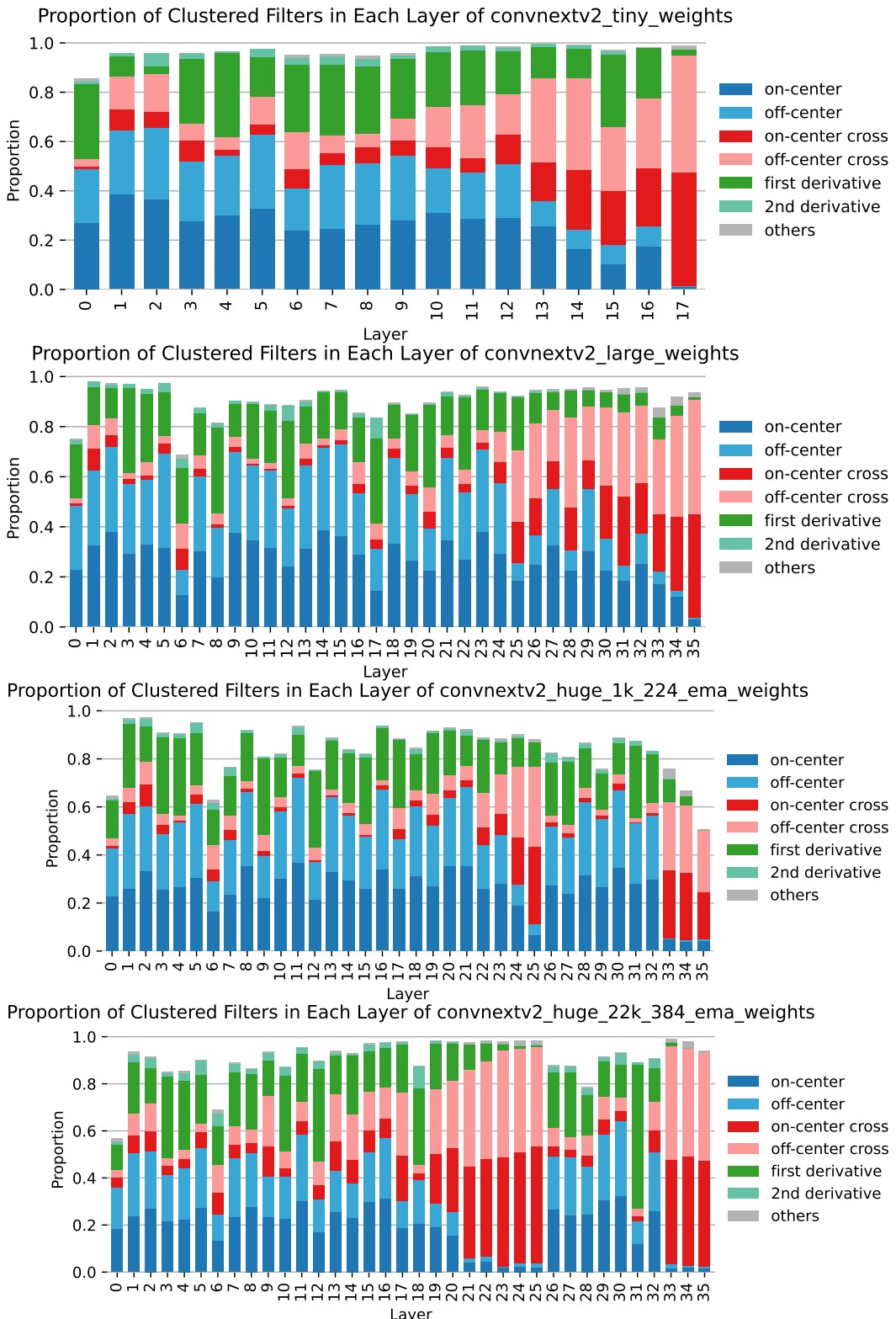

Figure 35: Barplot of relative cluster proportions across all Layers

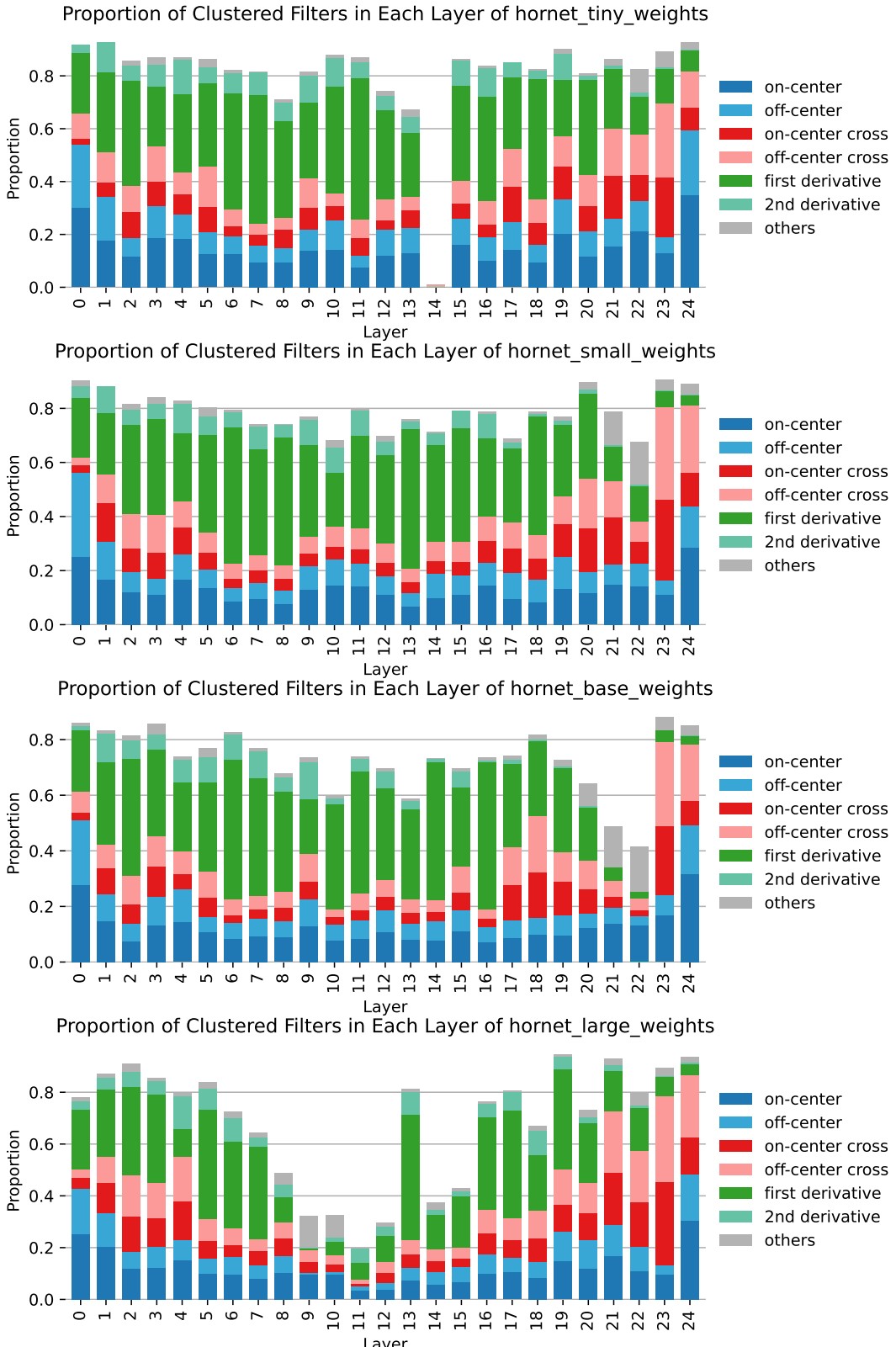

Figure 36: Barplot of relative cluster proportions across all Layers

Proportion of Clustered Filters in Each Layer of efficientnet_b0_weights

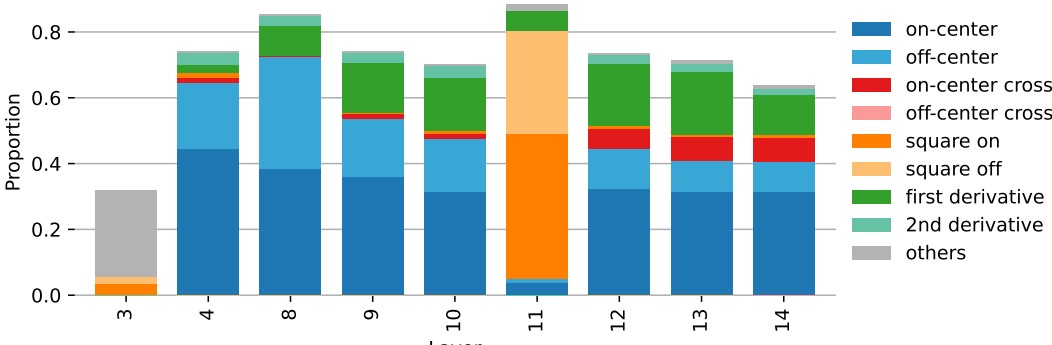

Proportion of Clustered Filters in Each Layer of efficientnet_b2_ra-bcdf34b7_weights

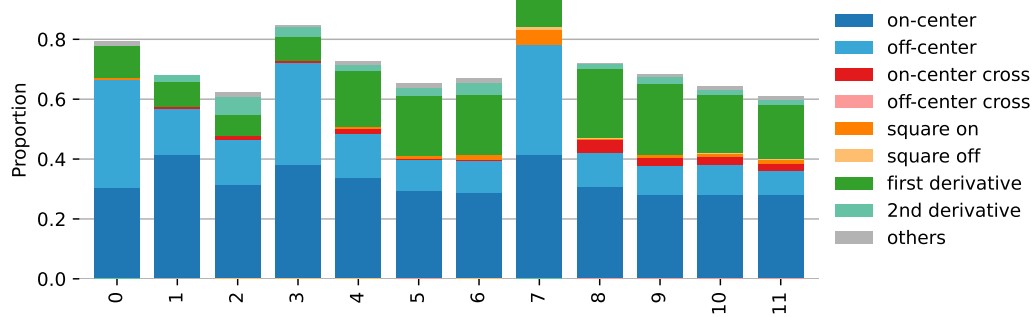

Proportion of Clustered Filters in Each Layer of efficientnet_b4_ra2_320-7eb33cd5_weights

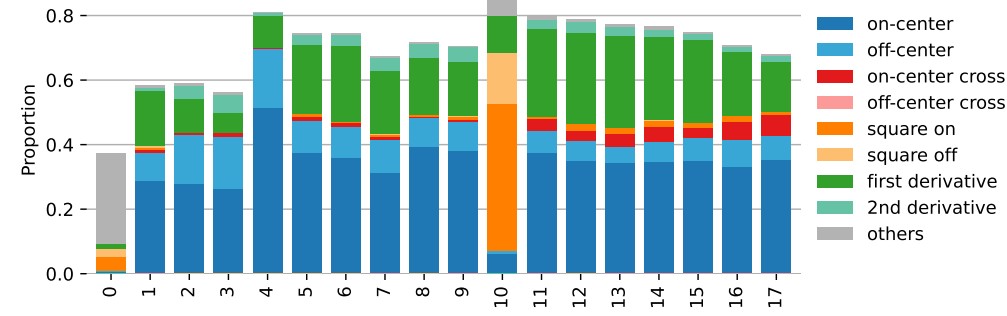

Proportion of Clustered Filters in Each Layer of tf_efficientnet_b6_ns-51548356_weights

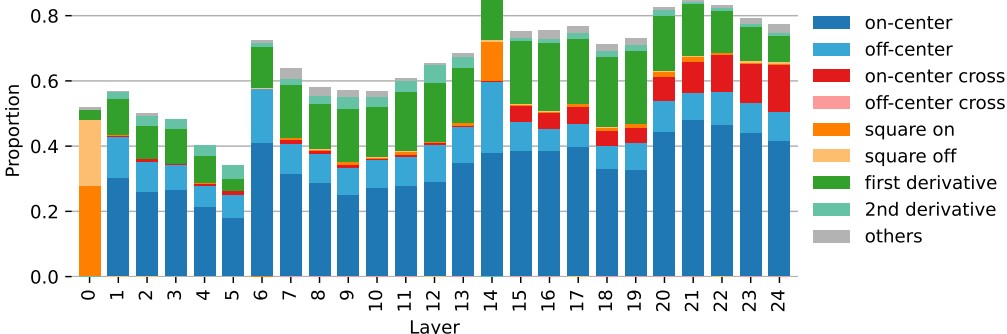

Figure 37: Barplot of relative cluster proportions across all Layers

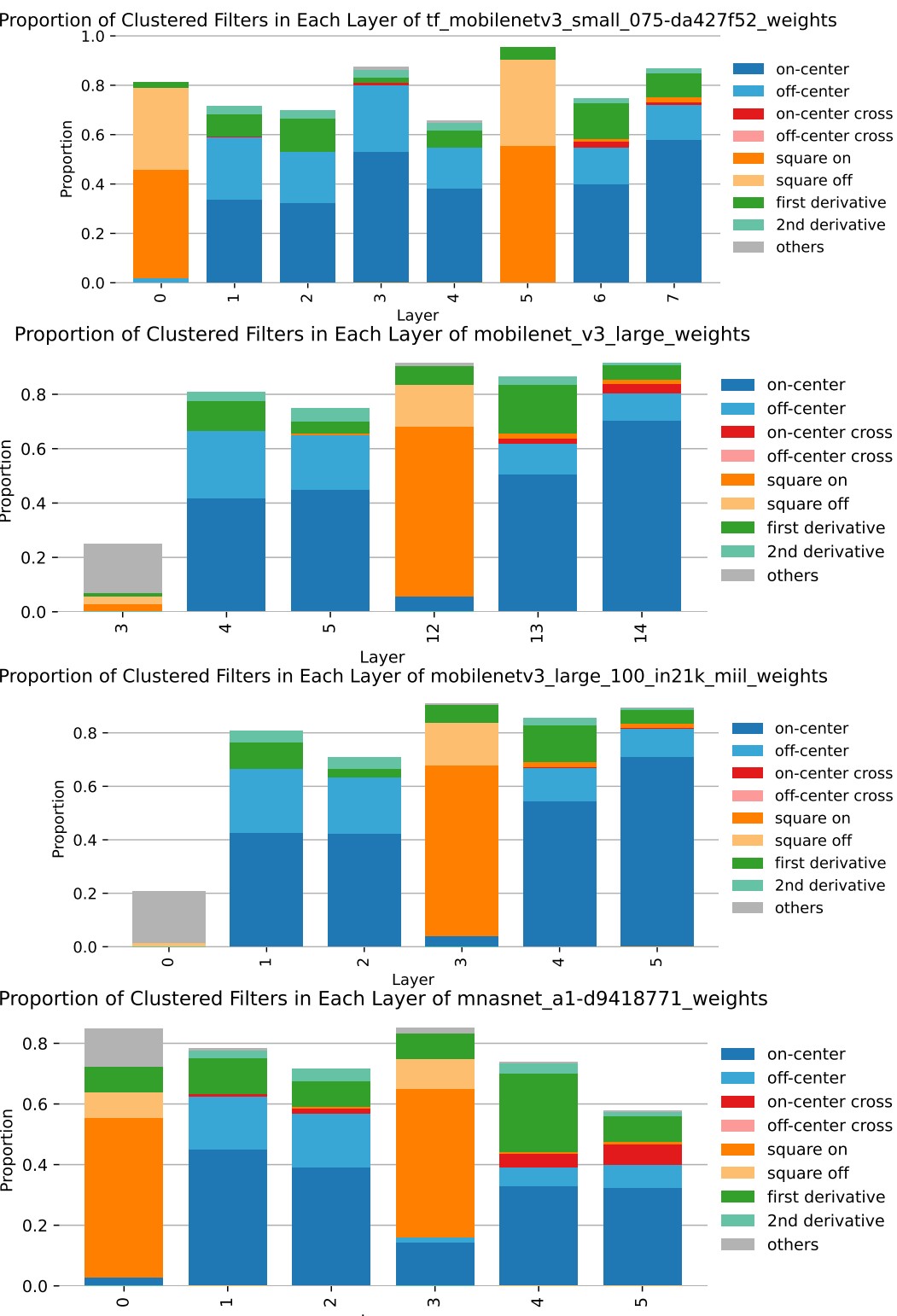

Figure 38: Barplot of relative cluster proportions across all Layers

# N    K-MEANS CLUSTERING OF MOBILENETV3 LARGE $3 \times 3$ KERNELS

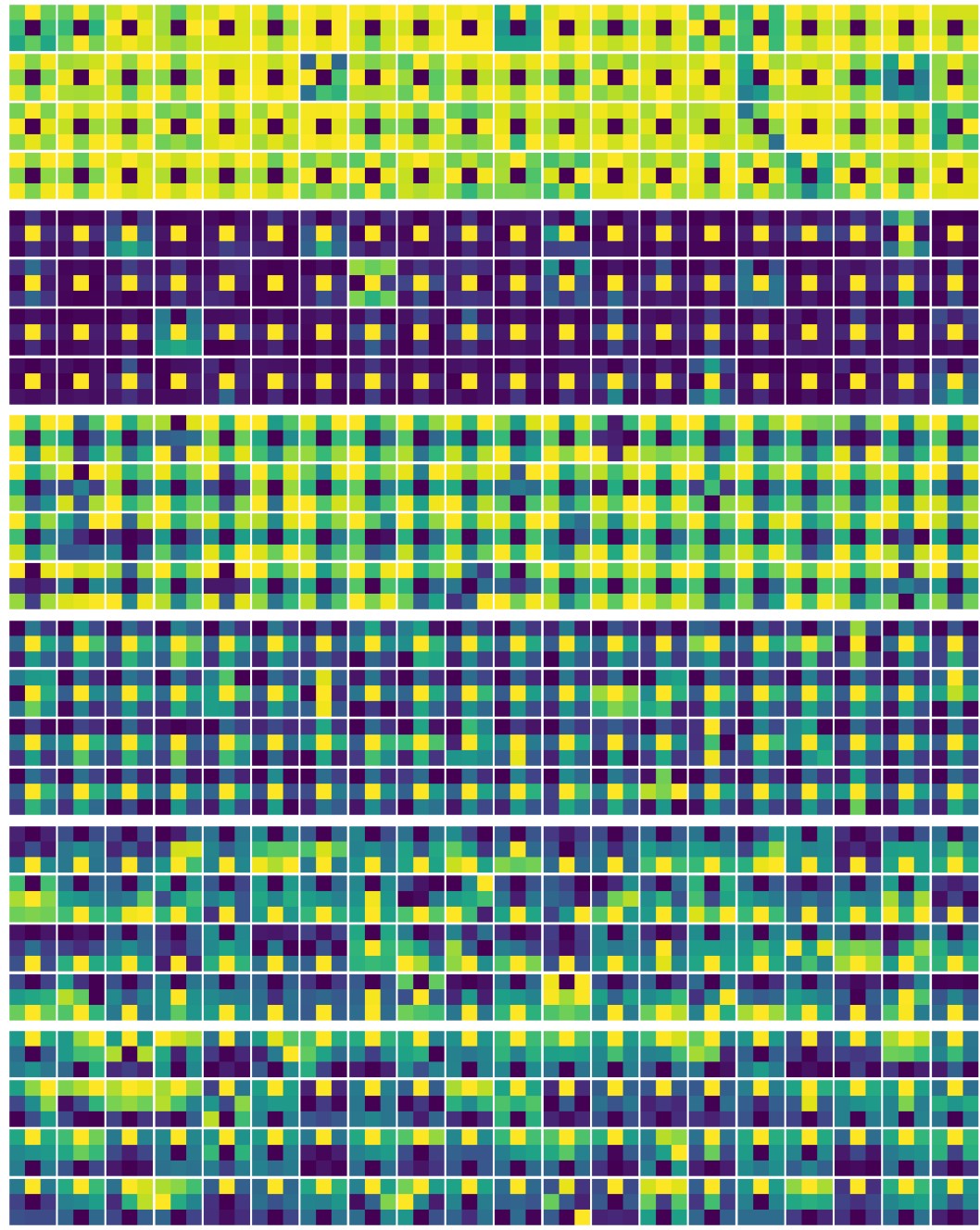

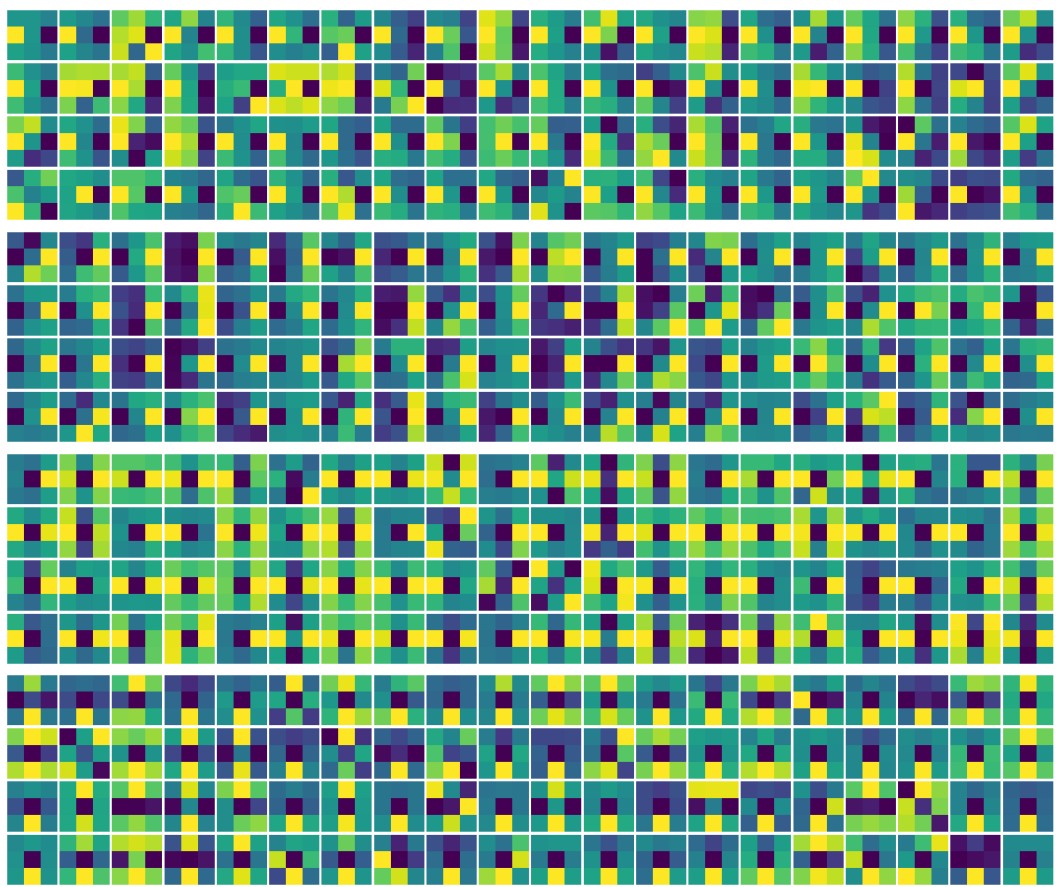

