# OpenReview forum: "Unveiling the Unseen: Identifiable Clusters in Trained Depthwise Convolutional Kernels"
_ICLR.cc/2024/Conference — ICLR 2024 poster_

### Official Review · Reviewer_9rPz · 2023-11-01

**Soundness:** 3 good
**Presentation:** 3 good
**Contribution:** 3 good
**Rating:** 8
**Confidence:** 4

**Summary:**

The authors examine trained filters in depthwise-separable CNNs in order to shed light into learn representation in this type of CNNs.
Preliminary visual inspection of those filters shows surprising regularity in their patterns and consistency across layers.
After an initial PCA-based exploration of those patterns, the authors devised an auto-encoder based clustering to identify and visualize those patterns, which correspond to DoG functions and their 1st- and 2nd-order derivatives.
The authors analyzed those patterns in various depth-wise CNNs trained on ImageNet-1k and ImageNet-22k to study the prevalence of corresponding filters at different layers.

**Strengths:**

+ A novel analysis of CNN filters that focuses specifically on depthwise-separable CNNs.
+ The analysis results seem novel, robust, and insightful. They demonstrate an important property of depthwise convolution that might make it better suited for future CNN-based architectures than standard convolution.
+ The insights can help improve the design of depthwise-separable CNNs e.g. w.r.t. kernel size, number of filters per layer.
+ The insights offer useful parallels to neuroscientific DoG-derivatives models.

**Weaknesses:**

These are not major weaknesses, but suggestions to generalize and strengthen the results:
- ImageNet is object centric. It might be useful to analyze the patterns in a model trained on a scene classification (e.g. Places-365) or other scene-centric tasks (e.g. semantic segmentation of CityScapes) in order to find if new patterns emerge.
  - To appreciate the impact of the dataset, check the examples of the 1st-layer filters reported in https://arxiv.org/abs/2204.08601
- Have you considered the impact of other elements of convolution? For example, padding in intermediate layers can skew the filters learned if applied in a one-sided way as reported in https://openreview.net/forum?id=m1CD7tPubNy
  - I think padding-induced skewness might offer explanation for the observed On-Square and Off-Square patterns. They might simply be skewed versions of the On-Center and Off-Center patterns.
- Have you considered the impact of global pooling? It was shown to have a profound impact of the weights learned and the patterns that emerge within them https://distill.pub/2020/circuits/weight-banding/
- It would be insightful to compare the reported findings with Deep Continuous Networks (Tomen et al., ICML 2021), where Gaussian derivative functions also emerge in the learned filters and the authors also draw similar biological parallels.


Language issues:
- Use \citet instead of \cite to avoid interference with the text (e.g. "natural images Krizhevsky et al. (2012)")
- of both regular and DSCs => regular CNNs and DSCs
- . in => In
- We focus on [...[, apply [...] => and apply
- The samples in each category, exhibit => no comma
- The On-Centre and Off-Centre clusters, show => no comma
- the recurring patterns we uncovered, arise => no comma
- we demonstrated the predominant => that predominant
- KMEANS (in appendix A.2) => K-Means

**Questions:**

See above

---

> ### Author Response · Authors · 2023-11-20
>
> We greatly appreciate your positive evaluation and insightful suggestions. Here are our detailed responses to the points you've raised.
>
> Weaknesses:
>
> **1.** Thank you for recommending the exploration of models trained on different datasets. In response, we sought DS-CNN models with publicly available weights trained on diverse datasets. We found the InternImage [1] models trained on the [ADE20K](https://groups.csail.mit.edu/vision/datasets/ADE20K/) Semantic Segmentation task. The clusters of the trained filters of the InternImage-T can be found [here](https://ibb.co/FWYDJg9). Notably, this model employs 3x3 kernels and uses Deformable depthwise convolutions. Despite the model's use of deformable convolutions and its training on semantic segmentation with scene-centric images, the filters it learns are quite similar to those of standard depthwise convolutions trained on ImageNet for classification. This similarity is evident when comparing them to the 3x3 MobileNet kernels detailed in our Appendix. This finding underscores the generalizability of our observations across different datasets and tasks.
>
>
> **2.** We agree with the reviewer that the squared filters may be a skewed version of DoGs. However, our analysis suggests that one-sided padding is not a primary factor in the emergence of these patterns. To further investigate, we trained two new instances of Efficientnet-B4 with different random seeds, detailed in Section D of the appendix. We consistently observed the same patterns in layer 10 with 5*5 kernels of this model, which is notably the first layer in a block of eight layers following a six-layer block. Additionally, we found a higher proportion of clustered filters in the first layer of the preceding block. This pattern recurrence in the initial layers of different blocks suggests that the position of a layer within the network's architecture, specifically being at the beginning of a block, has a more pronounced influence on the filters learned in that layer. This finding points towards architectural design, rather than padding techniques, as a more significant factor in the formation of these filter patterns.
>
> **3.** Thank you for bringing up the impact of global pooling on the learning of weights. The patterns observed in DS-CNNs appear consistently across all layers, not just before global pooling. This pervasive presence of patterns in DS-CNNs, irrespective of the layer, suggests that their formation is less influenced by global pooling compared to traditional CNNs.
>
>
> **4.** Thank you for the suggestion to compare our findings with those from this work. Their research, inspired by neuroscience, involves training continuous depth CNNs with filters that are a weighted sum of DoG and its derivatives, with the sum's weights being trainable. Recognizing the relevance of this work, we have added it to the related work section in our revised manuscript.
>
>
> Language Issues:
>
> Thank you very much for noticing these issues, we revised our manuscript and fixed them.
>
> [1] InternImage: Exploring Large-Scale Vision Foundation Models with Deformable Convolutions. Wang, W., Dai, J., Chen, Z., Huang, Z., Li, Z., Zhu, X., Hu, X., Lu, T., Lu, L., Li, H., et al. (2022).

---

### Official Review · Reviewer_GKik · 2023-11-01

**Soundness:** 4 excellent
**Presentation:** 4 excellent
**Contribution:** 4 excellent
**Rating:** 8
**Confidence:** 2

**Summary:**

The work posits that convolutional architectures that have depth-wise separable convolution layers learn a few characteristically Gabor-like filters, which is a widely intuited concept for CNNs in general but never really proven in any sense. They demonstrate analytically that this observation spans a zoo of models of various origins with various characteristics and is not a happenstance. They also show that this is not only a phenomenon that shows up in the initial layers of the network as was already demonstrated by several works prior, but in deeper layers of the network as well. However, the authors stop short of drawing strong conclusions out of their novel and interesting observations.

**Strengths:**

The work is not completely, original in its thesis - scientists have long intuited the nature of convolutional features from the perspective of biological vision and compositionality. There are several well-known images in well-read pieces of work from the original AlexNet paper to the VGG paper and so on, that present the filters in their first layer and their seeming relationships to Fourier or Gabor basis functions, lest one forget the image and its mystery in the AlexNet paper that shows that one GPU learnt color while the other learnt grayscale. What is original in this work is that the authors have noticed that for depth-wise convolutional networks, these features remain Gabor-like much deeper into the network. Not only do they demonstrate that, they also show that there are only a handful of these features and they are all lower-order filters as well. Stronger yet is the correspondence the authors have shown that the more performant the network, the more stronger these observations hold. This observation raises several questions regarding filter complexity, model complexity and regularization for convolutional networks and its relationship to the the order of these filters. This work is strong is so far as it is set to drive other future works in this area in the direction of a potential canonical and foundational vision model.

**Weaknesses:**

As strong as these observations are, the reviewer feels that this work is incomplete. The following are the reasons why:

1. The regularization argument: The authors note in several times including in their conclusions that the more lower-order and identifiable the features in a network are, the higher the generalization performance. One leads to wonder if these lower-oder DoG-like or Gabor-like filters have a regularization effect on the networks leading to higher generalization performance. This tantalizing hint should be explored further, even unto discovering a regularization process.

2. Multiple pre-trained models for same architecture: The authors used pre-trained models on Imagenet for their analysis. There is a need here to make these conclusions stronger, that this effect is not only present on these particular instance of these models but are an inherent property of the architecture. To do so, further analysis need to be performed to demonstrate these observations over multiple training runs on imagenet and also over other datasets of both natural images and also images from other domains. This would make the observations stronger.

3. Initialization/fine-tuning argument: In many a work on fine-tuning strategies such as in distillation, knowledge transfer, continual/incremental learning, observations has been made that so-called higher-frequency features often change and lower-frequency features remain unchanged. It also been noted in several works that fine-tuning mobilenets are harder than fine-tuning "general purpose" networks such as ResNets. This is largely the reason why most commercial use-cases involving fine-tuning convolutional networks use non depth-wise convolutional networks as their backbones. This here is a good stage to test this hypothesis and provide analytical evidence for or against the idea that networks that learn lower-order features do not change much during fine-tuning.

4. Well-fit, over-fit, under-fit models: These observations are made at the end checkpoint of these models. There arises a temporal question of when in the process of training do these filters form. It was noted by Fergus et al., that the first layer filters on convolutional networks take the shapes of these DoG-like or Gabor-like filters very early on in training and the latter stages of training only change the high-frequency filters. The author's workbench is the ideal setup to test this hypothesis. Accompanying this question is also that of over-fit and under-fit networks. High-frequency filters are often an indication of overfitting in non-neural network settings, such as in dictionary learning for instance. The authors could deliberately overfit some of these networks and demonstrate the emergence of more of their "other" category of filters.

The reviewer understands that ICLR has a page limit on its work, but also would like to recognize that while this work is strong, there are a lot of actionable directions that could make stronger and more conclusive statements on top of the already well-done analysis.

**Questions:**

1. The obvious unanswered question from the paper is the mystery layer 10 of EfficientNet-b4 and b6, and layer 11 of b2. This would ideally be the case where multiple checkpoints be needed to verify that this is not just a quirk of the random seed of training this model.

2.  What is happening in the layer 14 of hornet tiny?

3. Is it possible to create a graph of the number of identifiable filters by your clustering algorithm v. the generalization of the network on a standard test dataset? This would further bolster the authors' argument that better networks have more simpler features.

---

> ### Author Response · Authors · 2023-11-20
> **Response to Weaknesses**
>
> Thank you very much for your positive assessment of our work, detailed review, and constructive suggestions. We believe we can address all the issues to make a stronger paper. Below are the detailed responses to your feedback.
>
> Weaknesses:
>
> **1.**
> We agree that a potential outcome of our findings can be in developing novel initialization and regularization techniques. An initial experiment, detailed in Section F of the appendix, shows promising results where DoG filter initialization not only enhances accuracy but also increases the number of classified filters. While preliminary, this suggests potential for further improvement. With respect to regularization, we are looking into imposing low-order DoG constraints by employing our classification technique as a distance to be added to the loss function on DS-CNNs. However, this requires more thought and extensive experimentation.
>
> **2.**
> We appreciate the reviewer's insightful suggestion. In response, we have undertaken two additional analyses. Firstly, following your Question 1, we retrained the efficientnet_b4 model twice using different random seeds. The findings from these trainings are detailed in the appendix, Section D. Our observations showed that even with varying random seeds, the layer proportions remained consistent with the original model as released by PyTorch. This strengthens the hypothesis that these filter proportions are an inherent characteristic of the architecture, rather than a product of specific training instances. Secondly, we included Figure 11 in our revised paper, which displays the total proportions of clusters across all layers in different models. This revealed a compelling trend: models from the same family tend to learn similar filter proportions, irrespective of their size or training data.
>
> **3.**
> Our experiment on initialization with low-order DoGs supports the observations mentioned. We found that the proportions of filters in each low-order DoG cluster remained largely consistent even after 50 epochs of training the model (Please see Figure 23 in Section F of the appendix in our revised manuscript.)
>
> **4.**
> Thank you for your valuable suggestions.
>
> For under-fit models, we monitored the evolution of filters in the Convmixer768-32 model throughout its training, from the initial state up to epoch 50. This duration is shorter than the original model's 300-epoch training, thus representing an under-fit scenario. Our observations indicate a gradual development of DoG patterns, beginning from the earliest layers. Notably, the on-center DoG pattern is the first to emerge. These observations are detailed in Section G of the appendix in our revised manuscript.
>
> Regarding over-fit models, in line with your suggestion, we attempted to intentionally overfit the Convmixer768-32 model. We achieved this by training the model on a randomly selected 10-class subset of ImageNet for 200 epochs without any data augmentation. The outcome, detailed in Section H of the appendix and illustrated in Figure 37, revealed that the filters in the overfitted model lacked discernible patterns, appearing random. This result aligns with your prediction and supports our hypothesis that the emergence of DoG family patterns is linked to learning more general patterns, and hence the generalization capabilities of the models.

---

> ### Author Response · Authors · 2023-11-20
> **Response to the Questions**
>
> **1.**
> To further investigate this, we trained two new instances of Efficientnet-B4 with different random seeds, as mentioned above. We consistently observed the same patterns in the 10th 5*5 layer of this model, which is notably the first layer in a block of eight layers following a six-layer block. Additionally, we found a higher proportion of clustered filters in the first layer of the preceding block. This pattern recurrence in the initial layers of different blocks suggests that the position of a layer within the network's architecture, specifically at the beginning of a block, has a more pronounced influence on the filters learned in that layer. However, we still do not know what might be the exact reason behind it.
>
> **2.**
> We currently do not have a definitive explanation for layer 14's behavior. Upon closer examination of its filters, we observed that they lack any clear, discernible patterns. To provide a better understanding, we have included samples of filters from this layer, as well as from the adjacent layers, in Section E of the appendix. This comparative illustration highlights the unique characteristics of layer 14 in contrast to its surrounding layers.
>
> **3.** We tested the models ConvnextV2 Huge and ConvnextV2 Large, both trained on ImageNet-1k and ImageNet-22k, on the ImageNetV2 test sets (link:https://imagenetv2.org/). The table below summarizes the findings, illustrating the relationship between the proportion of identifiable filters (as clustered by our algorithm) and the generalization performance of the network on the new test datasets:
>
> | Model | Training Data | Clustered Filters (%) | ImageNet  (%) | ImageNetV2 Top Images  (%) | ImageNetV2 Matched Frequency (%) | ImageNetV2 Threshold 0.7 (%) |
> |-------|---------------|-----------------------|-------------------|--------------------------------|------------------------------------|------------------------------|
> | ConvnextV2 Huge | 1k | 82.08 | 86.3 | 87.130 | 77.790 | 84.200 |
> | ConvnextV2 Huge | 22k | 92.41 | 88.7 | 88.060 | 80.310 | 85.750 |
> | ConvnextV2 Large | 1k | 90.82 | 84.3 | 86.250 | 76.370 | 83.030 |
> | ConvnextV2 Large | 22k | 96.63 | 86.6 | 87.200 | 77.950 | 84.270 |
>
>
> We hope that our revision and additional experiments we have provided in response to your comments will increase the confidence in your assessment of our work. We believe these enhancements significantly strengthen our submission.

---

### Official Review · Reviewer_hU9E · 2023-11-01

**Soundness:** 3 good
**Presentation:** 3 good
**Contribution:** 2 fair
**Rating:** 3
**Confidence:** 5

**Summary:**

This paper performs an extensive study on how the depthwise convolution (dwconv) kernels in the depthwise-convolution-employed convolutional neural networks (this paper calls them DS-CNNs)  look after training on an image classification task. The authors employ the ImageNet pre-trained DS-CNNs such as ConvNeXts, EfficientNets, and MobileNets for the analysis. The main analysis tools are unsupervised clusterings via autoencoder and k-means clustering, and the clustered kernels are seemingly similar in shape to those of DOG filters (including the first and second derivatives). The authors report several qualitative results of clustered kernel shapes from different architectures and the histogram reflecting the portions of the representative kernel shapes.

**Strengths:**

- To my knowledge, this paper first analyzes the depthwise convolution.
- The idea of analyzing DS-CNNs by visualizing the weights that are clustered to representative shapes is interesting.
- This paper is easy to follow.

**Weaknesses:**

- The analysis is thought-provoking, but its significance is not fully realized. The ultimate goal of the analysis remains unclear to me. The paper effectively renders the clustered kernel in shapes akin to DOG or its derivatives, but it does not progress towards a more meaningful conclusion. This reviewer feels like there is a missed opportunity to extract and impart lessons, such as a connection to new network design principles for CNNs, which the analysis could potentially offer, yet such insights are not provided in the end.

- This reviewer is questioning the necessity for the advanced and training-needed autoencoder-like denoising/clustering technique in the analysis when K-means clustering appears to suffice for simply examining the trends of the learned kernels, as shown in the authors’ treatment of MobileNetV3. It is speculated that the denoising effect may smooth the actual filter's learned patterns, so the results may not be convincing. Furthermore, while the autoencoder is presumed to be more suitable for this analysis, a clear justification for this expectation is needed to validate its use as an analytical tool.

- The introduced autoencoder-based analysis presumably gives more sophisticated results over K-means-like methods. However, there is speculation that the authors are employing a tool that aligns more closely with the desired DoG patterns. Furthermore, such learning-based methods would give underfitted results when training is not successfully performed.

**Questions:**

- I am still not convinced that the patterns of the learned kernels follow DoG shapes, because 7x7 kernels are still small. There are the DS-CNNs that employed much larger kernel sizes, if the authors would like to claim the filter shape is restricted to DOG or its derivatives, the authors should try to verify the claim on the models consisting of larger kernels [1, 2].
  - [1] Scaling Up Your Kernels to 31x31: Revisiting Large Kernel Design in CNNs, CVPR 2022
  - [2] More ConvNets in the 2020s: Scaling up Kernels Beyond 51x51 using Sparsity, ICLR 2023

-  What is the primary goal of computing the filter classification score in Table 1? and what does it imply when the filter classification accuracy is high?

- Table 1 repots wrong ConvNeXtV2_tiny's top-1 accuracy on ImageNet-1k, which is not 84.9%. Please refer to the original manuscript.

Pre-rebuttal comments)
This reviewer finds analyzing the depthwise convolution and revealing common DoG-like patterns from DS-CNNs, which gives some insights. However, the impact of the analysis itself and further use cases remain unclear; the potential impact is also problematic and uncertain. Overall, the reviewer believes the paper does not reach the publication standards of ICLR in its present state, I would like to see the other reviewers' comments and the authors' responses.

---

> ### Author Response · Authors · 2023-11-20
> **Response to Weaknesses**
>
> We are grateful for your detailed review. In the following response, we address each of the concerns you have highlighted.
>
> Weaknesses:
>
> 1. As Reviewer GKik pointed out, the originality of this work consists in noticing that only a handful of DoG-like filters are learned in all subsequent layers of the DS-CNNs. The ultimate goal and the significance of this paper is to show, for the first time, that on average, more than 80-90% of the state-of-the-art DS-CNN filters are explainable by a single function, which is DoG. It is our hope that this paper will drive other future works in this area in the direction of a potential canonical and foundational vision model.
>
> 2. While K-means is good at revealing overall trends, it lacks the precision needed for our research goals. There are several reasons why k-means clustering is not functional for our case:
> - High-Dimensional Efficacy: Our method using an autoencoder outperforms K-means in clustering high-dimensional filters, a task where K-means struggles in our case.
> - Identifying Unclassified Filters: K-means cannot categorize filters as "unclassified," leading to a lot of noise, as evidenced in Section N of our appendix. In contrast, our autoencoder uses a loss to effectively identify and label unrecognized filters. For an example of unclassified filters, please see Section E of the appendix in the revised manuscript.
> - Consistent Criteria Across Models: The autoencoder provides a uniform clustering standard for all models, ensuring a fair comparison. K-means, however, treats each model separately with varying cluster centers, restricting consistent analysis.
> - Clustering in Underfit Models: K-means fails to effectively cluster in underfit models due to a shortage of complete filter convergence. Our autoencoder approach is robust enough to handle such scenarios, demonstrating its adaptability and clustering capability. For an example of underfitted models, please see Section G of the appendix in the revised manuscript.
>
> > It is speculated that the denoising effect may smooth the actual filter's learned patterns, so the results may not be convincing.
>
> We are not sure if we have understood the reviewer's concern properly, but to address the speculation we would like to clarify that all the clustered filters presented in our paper are original and directly extracted from the models and not reconstructed by the autoencoder. Our use of the autoencoder was exclusively for clustering purposes, without altering the filters.
>
>
>
> We hope our explanations have justified the use of autoencoders in our analysis. If there are still doubts regarding their superiority over K-means for this specific application, we welcome further discussion to clarify any remaining concerns.
>
> 3. We would like to clarify that the autoencoder’s training was unbiased and unsupervised, involving more than one million filters from diverse models. We employed a strict loss threshold for the autoencoder reconstructions to prevent underfitting and false-positive classifications. Had there been any bias in the autoencoder model, it would have resulted in a significant number of unrecognized filters in most models, which was not observed in our analysis. The accuracy of our classification is evidenced by the random cluster samples in Figures 5 and 6, and Figure 20 in the appendix further demonstrates our method's effectiveness in correctly identifying 'unrecognized' filters.

---

> ### Author Response · Authors · 2023-11-20
> **Response to the Questions**
>
> **1.**
> We thank the reviewer for suggesting larger kernel DS-CNNs. In response, we extended our investigation to include larger kernel DS-CNNs. Specifically, we examined the RepLKNet-XL model, referenced in [1], which utilizes 27x27 filters. Our observations confirmed the presence of similar DoG-shaped filters in this model as well. To illustrate this, we have included selected examples of these larger filters in Section I of the appendix in our revised manuscript. Despite their larger size, some of these filters also exhibit a central focus characteristic of DoG shapes and their derivatives, although they are much higher in dimension and have fewer proportions of these kernels compared to models with smaller kernels. Regarding the paper [2] you mentioned, it utilizes sparse 51x5 and 5x51 kernels, which differ from the square kernels we focused on in our study.
>
> The DoG filters and their derivatives have long been proposed to model the biological visual system, as well as their widespread application in traditional image processing. In this paper, we conducted an extensive analysis of the trained filters across several DS-CNN models, and we discovered that many of these filters are similar to the shape of DoGs and their derivatives, without any direct supervision to this effect. We interpret this high frequency of occurrence as more than mere coincidence. We hope our additional clarifications and the provided examples address your concerns. If the reviewer holds a possibility of this resemblance being coincidental, we are more than willing to engage in further discussion of this topic.
>
> **2.** The filter classification score in Table 1 represents the percentage of filters in each model that we categorized into a main DoG filter family. Some filters, marked as 'not classified', showed higher reconstruction loss and were more cluttered, although some still resembled our main classes. A higher classification score indicates a greater proportion of filters in that model were successfully classified.
>
> **3.** Thank you very much for pointing this out. We have corrected this table in our revised manuscript.
>
> We hope that our response adequately addresses the reviewer's points and hope it positively influences their assessment of our work.

---

### Official Review · Reviewer_Jbbg · 2023-11-09

**Soundness:** 3 good
**Presentation:** 3 good
**Contribution:** 3 good
**Rating:** 6
**Confidence:** 4

**Summary:**

This paper proposes a study on the nature of the learned convolutional filters of various CNN architectures. They show that the kernels learned by depthwise-separable convolutions (DSC) are similar to DoG-derivatives filters, which draws connections with biologically inspired models.

**Strengths:**

1) The paper offers a very insightful and novel view on the fact that DSC kernels learn DoG filters, which is to the best of my knowledge a new observation for CNN architectures. The visualizations, especially Figure 2, are very informative and clearly show this property of DSC.

2) The clustering of kernels approach shows that the learned kernels which are DoG filters actually form a basis of DoG filters with specific shapes that the authors identify and show in Figure 5. This observation shows that DSC are more similar to biological models than regular convolutions which could inspire future CNN architectures.

3) The claims of the paper are validated on a wide variety of models and many visualizations are presented, including in-depth study of each layer of each model and a lot of examples of kernels.

**Weaknesses:**

1) The authors suggest that better understanding the nature of what CNNs learn, and in this particular case the fact that DSC learn DoG-filters, would lead to architectural improvements or new architectures. However no actual concrete improvements or suggestions are proposed in the paper.

2) The observations are interesting but a more interpretable model is not necessarily correlated to a better performance of the model, for example, vision transformers are the most popular architecture nowadays, they don’t learn filters and are not inspired by biology. A more in-depth study of the performance on various tasks of models with DS-conv against models without would have been a good addition. For example, a study on the differences between ResNet and ResNeXt/ConvNeXt families. Or with a specific architecture by only replacing regular convolution with DSC.

3) The paper claims that DSC learns a basis of DoG filters, which is highlighted by the visualizations and the clustering approach, but it might be the case that ResNet models and other models using standard convolutions also learn a basis of other functions/filters that are less interpretable, but still efficient for the final performance of the model. It is not clear that interpretability is a key advantage.

4) The interpretation of what is the role of each ‘basis vectors’ DoG filter (such as “off-center” or “off-center cross”) is doing would help clarify what is the advantage for a CNN model to learn such filters.

5) DS, DC, and DSC notations are used interchangeably throughout the paper and it is not clear if they mean the exact same thing or not. Could you clarify or use the same term for consistency ?

6) Most captions of figures are not self-contained and very unclear, especially on describing how the visualizations are obtained and what conclusion should be drawn from the figure.

**Questions:**

1) In Figure 7, layer 10 has a strong proportion of ‘square off’ kernels, do you have an intuition on why this is the case ?

2) Why do you need to use an autoencoder to learn kernel representations, could you not just use the kernels themself and use k-means ?

---

> ### Author Response · Authors · 2023-11-20
>
> We would like to thank you for the thorough review and for raising important points. Below, we address the points raised by you.
>
> Weaknesses:
> 1. As Reviewer GKik pointed out, the originality of this work consists of noticing that only a handful of DoG filters are learned in all subsequent layers of the DS-CNNs. It is our hope that this paper will drive other future works in this area in the direction of a potential canonical and foundational vision model.
> A potential outcome of our findings can be in developing novel initialization and regularization techniques. An initial experiment, detailed in Section F of the appendix, shows promising results where DoG filter initialization not only enhances accuracy but also increases the number of classified filters. While preliminary, this suggests potential for further improvement. With respect to regularization, we are looking into imposing low-order DoG constraints by employing our classification technique as a distance to be added to the loss function on DS-CNNs. However, this requires more thought and extensive experimentation.
>
> 2. We agree that performance is not necessarily related to interpretability. While we recognize the significance of performance comparisons, the aim of this paper was not to compare performance metrics across different architectures, like vision transformers, but just to deepen the understanding of DS-CNNs' structural features.
>
> 3. You rightly point out that models like ResNets, using standard convolutions, may learn a basis of functions or filters that are less interpretable but still efficient. The goal of this paper was not to claim that interpretability is an outright advantage in performance but to contribute to the understanding of how different architectures process information. We believe this adds valuable perspective to the ongoing discourse on neural network design and function.
>
> 4. Our research identifies the presence of DoG and derivative filters in the deeper layers of DS-CNNs, beyond their common application in image processing. DoGs are known for edge detection in images, while derivative filters track changes in brightness. However, in DS-CNNs, these filters operate on more abstract feature maps in deeper layers. Further study is required to determine the specific roles and advantages of these filters at each layer of a CNN model.
>
> 5. Thank you for pointing out the inconsistency in our terminology. To clarify, "DSC" refers specifically to depthwise separable convolutions, a term encompassing both depthwise convolutions (abbreviated as "DC") and pointwise convolutions. In response to your feedback, we have revised the manuscript to ensure consistent and clear use of these terms throughout.
>
> 6. Thank you for your valuable feedback. We have updated all figure captions to be more self-explanatory and clear. We would be grateful if you could let us know whether these revisions meet your expectations or if further improvements are needed.
>
>
> Questions:
>
> 1. To further investigate this, we trained two new instances of Efficientnet-B4 with different random seeds, detailed in Section D of the appendix. We consistently observed the same patterns in the 10th 5*5 layer of this model, which is notably the first layer in a block of eight layers following a six-layer block. Additionally, we found a higher proportion of clustered filters in the first layer of the preceding block. This pattern recurrence in the initial layers of different blocks suggests that the position of a layer within the network's architecture, specifically at the beginning of a block, has a pronounced influence on the filters learned in that layer. However, we still do not know what might be the exact reason behind it.
>
> 2. There are several reasons why k-means clustering is not functional for our case:
>
> - High-Dimensional Efficacy: Our method using an autoencoder outperforms K-means in clustering high-dimensional filters, a task where K-means struggles in our case.
> - Identifying Unclassified Filters: K-means cannot categorize filters as "unclassified," leading to a lot of noise, as evidenced in Section N of our appendix. In contrast, our autoencoder uses a loss to effectively identify and label unrecognized filters. For an example of unclassified filters, please see Section E of the appendix in the revised manuscript.
> - Consistent Criteria Across Models: The autoencoder provides a uniform clustering standard for all models, ensuring a fair comparison. K-means, however, treats each model separately with varying cluster centers, restricting consistent analysis.
> - Clustering in Underfit Models: K-means fails to effectively cluster in underfit models due to a shortage of complete filter convergence. Our autoencoder approach is robust enough to handle such scenarios, demonstrating its adaptability and clustering capability. For an example of underfitted models, please see Section G of the appendix in the revised manuscript.

---

### Author Response · Authors · 2023-11-22
**Summary of the Revisions**

Dear reviewers,

We would like to thank you all again for reviewing our paper, for recognizing our work as the first to analyze and identify DoG-like patterns in all layers of DS-CNNs, and for noting its potential to open new research avenues and canonical CNN model development.

As we are getting closer to the end of the author-reviewer discussion period, we would like to summarize our revision to the manuscript, with the corresponding appendix sections.

1. **Figure 11 & Section J**: Our analysis revealed a significant uniformity in cluster proportions across different model families.
2. **Section F**: We conducted an initialization experiment using DoGs on the model with the least clustered filters. The results were promising, showing an increase in both accuracy and clustered filter rate.
3. **Sections D & E**: Addressing your queries, we examined anomalies in specific layers of the HorNet tiny model and the EfficientNet B4.
4. **Sections G & H**: Our investigation into underfit and overfit models uncovered a notable finding: overfit models demonstrate a lack of clear pattern.
5. In our study of the InternImage model, trained for scene image semantic segmentation, we found cluster formations similar to those in ImageNet-trained models. For visual reference, see [here](https://ibb.co/FWYDJg9).
6. **Section I**: Our exploration of the RepLKNet-XL model, especially its large 27*27 filters, revealed the presence of DoG-like filters.

Our revisions respond to the insightful queries raised by the Reviewers: gmsA (2, 3), hU9E (6), GKik (1, 2, 3, 5), and 9rPz (3, 5).

Sincerely,

Authors

---

### Meta-Review · Area_Chair_fZBc · 2023-12-08

**Metareview:**

This paper studies the learned convolutional filters of various CNN architectures. It shows that depthwise-separable convolutions (DSC) learn kernels similarly to DoG-derivatives filters, drawing connections with biologically inspired models. The authors employ unsupervised clustering methods and report qualitative results on the shape of clustered kernels from different architectures. It has been raised by reviewer hU9E that the paper “does not progress towards a more meaningful conclusion.” Indeed, this work does not go beyond making the primary analysis. At the same time, the same reviewer saluted the “thought-provoking” nature of experiments. In my view, the observation is attractive and well-executed. It should grant this paper acceptance at ICLR as a poster presentation.

**Justification For Why Not Higher Score:**

The analysis presented in this work is interesting, but as stated by reviewer hU9E the paper fails to show what would this analysis entail. Could this observation guide future design of convolutional architectures? If a parallel could be drawn with neuroscience, what would be the implications?

**Justification For Why Not Lower Score:**

The experiments presented in the paper are sound, and the observation is interesting and novel.

---

### Decision · Program_Chairs · 2024-01-16

Accept (poster)